# Allosteric enhancement of ORP1-mediated cholesterol transport by PI(4,5)P$_2$/PI(3,4)P$_2$

Jiangqing Dong[1], Ximing Du[2], Huan Wang[1], Jue Wang[3], Chang Lu[1], Xiang Chen[1], Zhiwen Zhu[1], Zhipu Luo[3], Li Yu[1], Andrew J. Brown[2], Hongyuan Yang [2] & Jia-Wei Wu[1,3]

Phosphatidylinositol phosphates (PIPs) and cholesterol are known to regulate the function of late endosomes and lysosomes (LELs), and ORP1L specifically localizes to LELs. Here, we show in vitro that ORP1 is a PI(4,5)P$_2$- or PI(3,4)P$_2$-dependent cholesterol transporter, but cannot transport any PIPs. In cells, both ORP1L and PI(3,4)P$_2$ are required for the efficient removal of cholesterol from LELs. Structures of the lipid-binding domain of ORP1 (ORP1-ORD) in complex with cholesterol or PI(4,5)P$_2$ display open conformations essential for ORP function. PI(4,5)P$_2$/PI(3,4)P$_2$ can facilitate ORP1-mediated cholesterol transport by promoting membrane targeting and cholesterol extraction. Thus, our work unveils a distinct mechanism by which PIPs may allosterically enhance OSBP/ORPs-mediated transport of major lipid species such as cholesterol.

[1] Beijing Advanced Innovation Center for Structural Biology, MOE Key Laboratory for Protein Science, Tsinghua-Peking Center for Life Sciences, School of Life Sciences, Tsinghua University, Beijing 100084, China. [2] School of Biotechnology and Biomolecular Sciences, the University of New South Wales, Sydney, NSW 2052, Australia. [3] Institute of Molecular Enzymology, Soochow University, Suzhou, Jiangsu 215123, China. These authors contributed equally: Jiangqing Dong, Ximing Du, Huan Wang. Correspondence and requests for materials should be addressed to H.Y. (email: h.rob.yang@unsw.edu.au) or to J.-W.W. (email: jiaweiwu@suda.edu.cn)

Late endosomes and lysosomes (LELs) are key degradative and recycling organelles, and they also serve as a critical nutrient sensing station where the mammalian target of rapamycin complex 1 (mTORC1) is activated[1,2]. Lipids within the limiting membrane of LELs, such as phosphatidylinositol (PI) phosphates (PIPs) and cholesterol, are known to regulate the movement, maturation and function of LELs[3–5]. Recent studies have also demonstrated that LEL cholesterol and PI 3,4-bisphosphate (PI(3,4)P$_2$) play critical roles in the activation or repression of mTORC1 activity, respectively[5,6]. Moreover, cholesterol is an essential component of mammalian membranes, where it plays important structural and functional roles to maintain normal cell function[7–9]. Besides de novo synthesis, mammalian cells can acquire cholesterol through receptor-mediated endocytosis of low-density lipoproteins (LDL). The endocytic pathway sorts and delivers LDL to LELs for the hydrolysis of cholesterol esters, and the released free cholesterol (LDL-C) can reach the limiting membrane of LELs through the collaborative actions of Niemann Pick C 1&2 (NPC1 and NPC2)[10,11]. However, how LEL membrane lipids are sensed, and how LDL-C leaves the limiting membrane of LELs and reaches other parts of the cell remains to be elucidated[12].

Growing evidence supports the notion that sterols and other lipids can be moved between membranes by lipid transfer proteins (LTPs) in a nonvesicular manner[13–15]. The oxysterol binding protein (OSBP) and OSBP-related proteins (ORP1-11) represent an evolutionarily-conserved family of LTPs in human[16,17]. Elegant work has demonstrated that OSBP can bind and transfer both cholesterol and PI 4-phosphate (PI4P) from apposing membranes, and a PI4P gradient is essential for OSBP to transfer sterol against a concentration gradient[18]. ORP1L is a unique ORP that specifically associates with NPC1-positive LELs[19], and has been established as a cholesterol sensor that regulates the microtubule-dependent movement of LELs, as well as the transport and fate of autolysosomes[3,4]. Recent studies also suggest that ORP1L may function as a cholesterol transporter that moves cholesterol between LEL and the ER. For example, ORP1L has been shown to transport cholesterol from LEL limiting membrane to the ER, and this transfer requires PI4P binding to ORP1L[20]. On the other hand, when LEL cholesterol is low, ORP1L facilitates cholesterol transfer from the ER to MVBs to support intralumenal vesicle formation[21]. Here, our results suggest that ORP1 is a unique sensor of lysosomal PI-bisphosphates, and that PI(4,5)P$_2$/PI(3,4)P$_2$ allosterically regulates cholesterol transport by ORP1.

## Results

### PI(3,4)P$_2$/PI(4,5)P$_2$ stimulates cholesterol transport by ORP1.
Members of the OSBP/ORP family vary in length: the short ORPs contain primarily a conserved OSBP-related domain (ORD) that can bind and transfer lipids such as sterols, PIPs or phosphatidylserine (PS), whereas the long ones often possess additional functional domains for ligand binding and/or membrane targeting. The *OSBPL1A* gene (homo sapiens) encodes two proteins: the long isoform ORP1L (residues 1–950) containing the ANK (ankyrin repeat) domain, the PH (Pleckstrin homology) domain, the FFAT (diphenylalanine in an acidic tract) motif and the ORD, and a short isoform ORP1S (residues 514–950) containing only the ORD (Fig. 1a and Supplementary Fig. 1). We expressed the ORP1S and a series of N-terminal deletion variants, among which only ORP1S and two truncations (residues 524–950 and 534–950) were soluble. Three assays were used to characterize the lipid binding specificity of ORP1-ORD[15,22,23] (Fig. 1b and Supplementary Fig. 2a). However, the fluorescent changes in these assays cannot differentiate whether the lipid is bound by ORP1-

ORD yet buried in liposome, or extracted and dissociated from liposome (see below). Hereafter, these assays were referred to as binding assays. In the sterol binding assay, the decrease in fluorescence resonance energy transfer (FRET) signal between DHE (dehydroergosterol, a natural fluorescent cholesterol analog[18]) and DNS-PE embedded in the liposomes allows the quantification of DHE bound by ORP1-ORD. The ORP1-ORD proteins could bind nearly 80% accessible DHE, similar to yeast Osh4p (Fig. 1b and Supplementary Fig. 2). In the PIP binding assay, the NBD-labeled PH$_{FAPP}$ (the PH domain of four-phosphate-adaptor protein 1) was used as the sensor for all PIPs[24], and the NBD signal would be quenched when NBD-PH$_{FAPP}$ is competed off by ORP/Osh proteins and dissociates from the liposome. PI4P binding was suggested to be a unifying feature of all ORP/Osh proteins[18,23,25], and ORP1-ORD indeed bound PI4P. Notably, the ORP1-ORD proteins could also bind all other mono-, di- and tri-phosphorylated PIPs. Therefore, ORP1-ORD could bind sterol and PIPs, but not PS, which is similar to yeast Osh4p but distinct from Osh6p (Fig. 1b and Supplementary Fig. 2b).

Based on the binding specificity, we hypothesized that ORP1-ORD might be an exchanger for cholesterol and PIPs like Osh4p and OSBP[15,18]. We first examined the sterol transport ability of ORP1-ORD with the donor liposomes (L$_A$) containing DHE and the acceptor liposomes (L$_B$) containing different PIPs or not (Fig. 1c and Supplementary Fig. 3a). The decrease in FRET signal between DHE and DNS-PE on L$_A$ was converted into the amount of DHE transported by ORP1-ORD. When the L$_B$ liposomes were free of any PIPs, the ORP1-ORD proteins could transport DHE/ cholesterol slowly. Interestingly, the DHE transport was markedly enhanced when the acceptor liposomes were supplemented with PI(3,4)P$_2$ or PI(4,5)P$_2$, while all other PIPs, including PI(3,5)P$_2$ and PI4P, had little stimulatory effect. The PI(4,5)P$_2$-stimulation on ORP1 (nearly 18-fold) is even more efficient than the PI4P-stimulation on Osh4p (Supplementary Fig. 3b). We also carried out the PIP transport assays with the L$_B$ liposomes as donor (containing different PIPs) and the L$_A$ liposomes as acceptor (containing DHE or not) (Fig. 1d and Supplementary Fig. 3c, d). When the sensor NBD-PH$_{FAPP}$ binds to PIPs embedded in the donor liposomes, the NBD fluorescent signal will be quenched due to FRET with Rhod-PE. Unexpectedly, addition of ORP1-ORD did not cause any changes in the NBD signal, regardless of the presence of DHE. Thus, ORP1-ORD can efficiently transport cholesterol in the presence of PI(3,4)P$_2$ or PI(4,5)P$_2$, but not PI4P. Strikingly, ORP1 could not transport any PIPs backward, distinct from OSBP and Osh4p that can exchange sterols for PI4P between membranes[15,18].

### ORP1L and PI(3,4)P$_2$ regulates endosomal cholesterol efflux.
To test whether ORP1 mediates cholesterol transport in vivo, we generated ORP1L single and ORP1L/S double knockout HeLa cells using CRISPR (Fig. 2a, b and Supplementary Fig. 4a, b). We first examined LEL cholesterol in normal and ORP1 knockout cells by staining the cells with filipin. While there appears to be increased filipin-positive endosomes in the knockout cells, the increase was less obvious than recently reported (Supplementary Fig. 4c)[20]. We treated cells with U18666A, which is known to disrupt NPC1 function and cause cholesterol to accumulate in LELs. Upon washing away U18666A, cholesterol exited LELs in a time-dependent manner (Supplementary Fig. 4d). Generally, there was a significant delay of cholesterol egress from LELs in the knockout cells compared to control cells, especially at 6 h post U18666A treatment (Fig. 2c, d). To further examine the transport of LDL-cholesterol from LELs to the endoplasmic reticulum (ER), we measured the incorporation of [14C]-oleate into cholesteryl

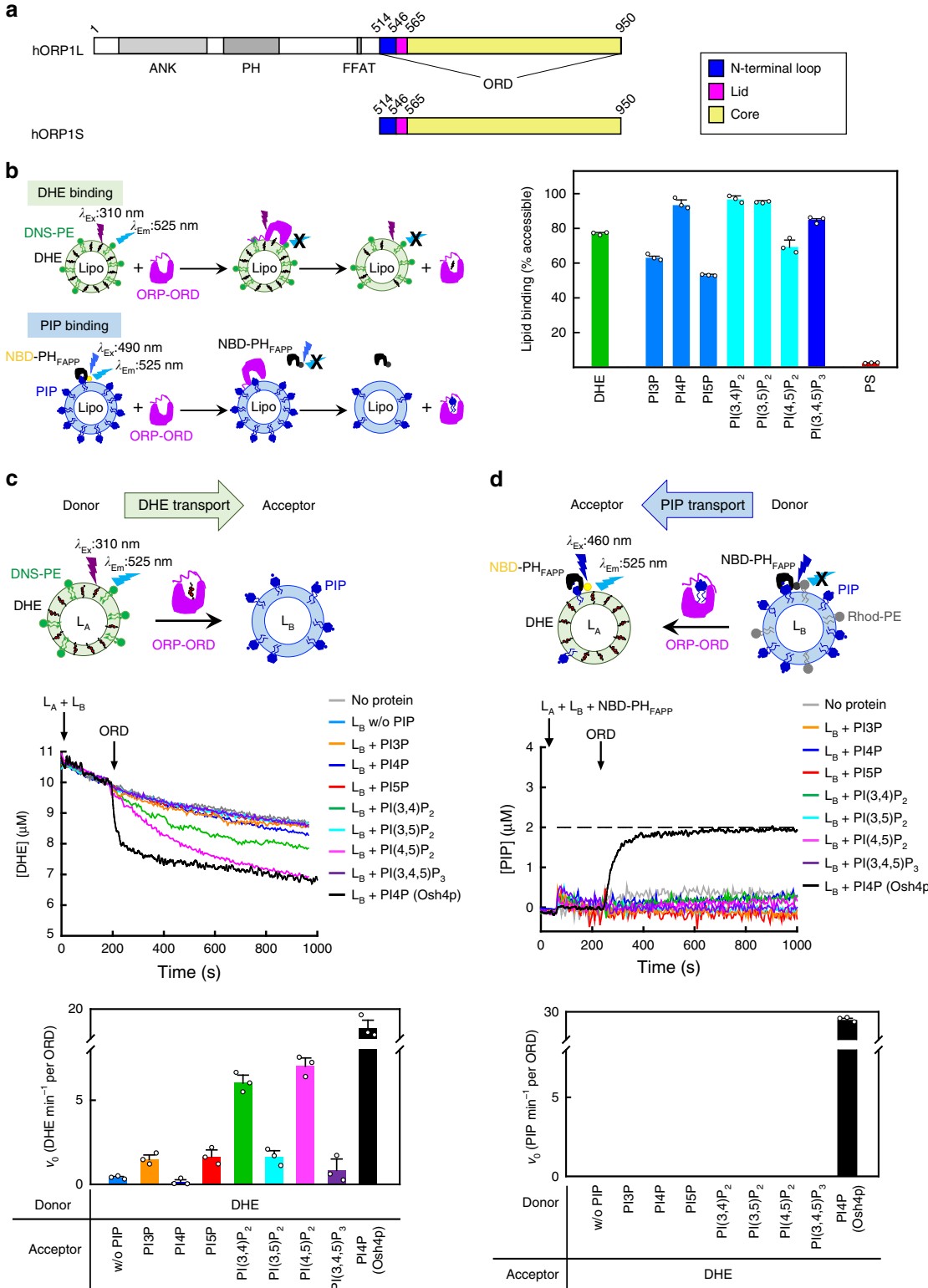

**Fig. 1** ORP1-ORD transfers cholesterol but not PIPs. **a** Schematic diagram of human ORP1L and ORP1S. The structural elements in ORD are colored as follows: N-terminal loop (blue), lid (magenta), core (yellow). The color scheme of ORP1-ORD is consistent throughout the schematic representations unless indicated. **b** Lipid-binding analyses for ORP1-ORD. **c** DHE transport assays without or with PIP in the acceptor liposomes. Shown at the bottom are the initial velocities of DHE transport subtracted by that without ORP1. **d** PIP transport assays with DHE-containing acceptor liposomes. (For all bar graphs, data are shown as mean ± s.e.m. (error bar), $n = 3$)

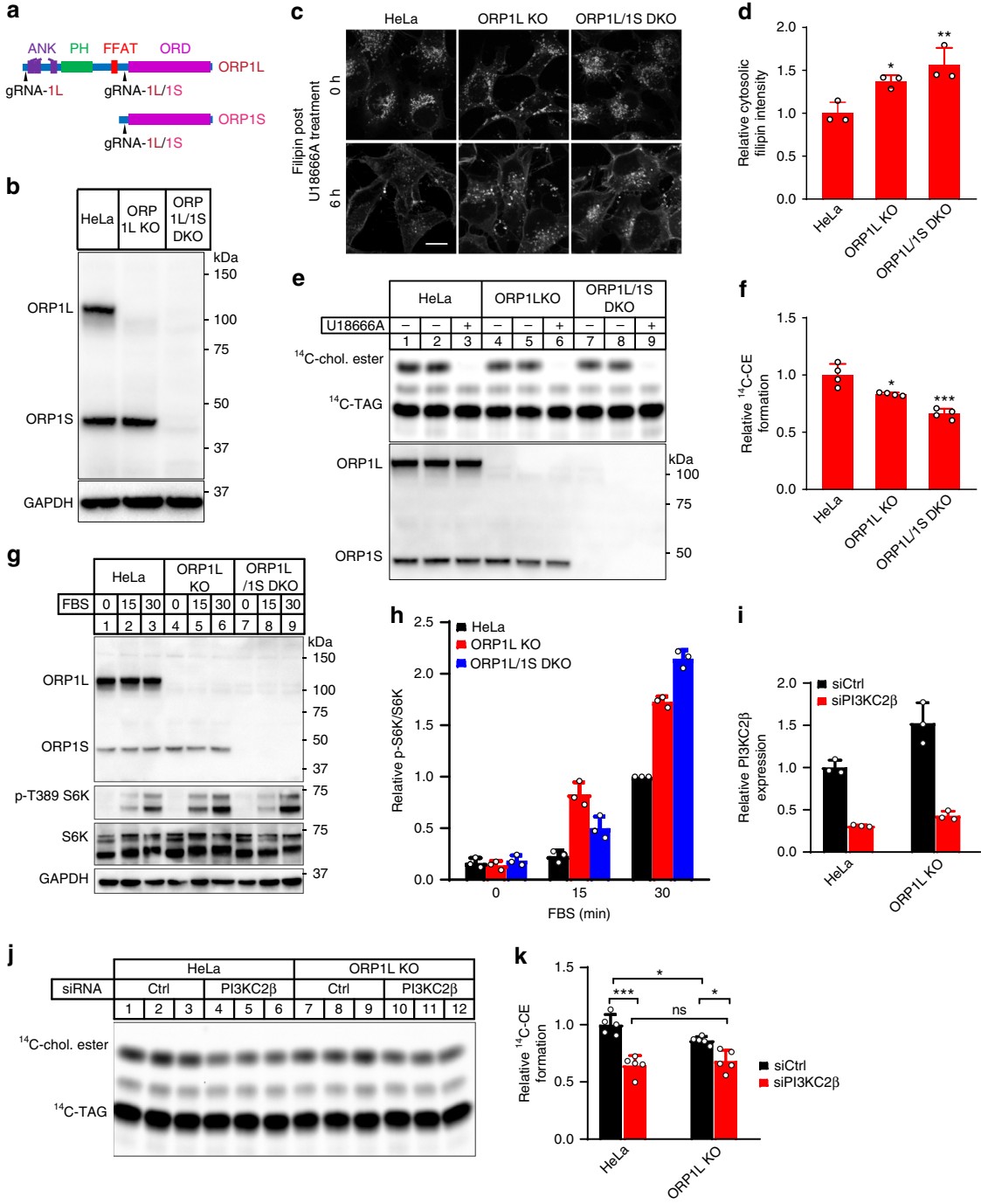

**Fig. 2** ORP1L is involved in endosomal cholesterol transport and signaling. **a** Guide RNA targeting sites for ORP1L and ORP1S. **b** Western blotting analysis of HeLa, ORP1L KO, and ORP1L/ORP1S DKO cells. **c** U18666A-mediated LEL cholesterol release assay in HeLa, ORP1L KO, and ORP1L/1S DKO cells (scale bar = 10 μm). **d** Quantitation of cytosolic filipin intensities of cells (6 h) in **c**. **e** Cholesterol esterification assay in HeLa, ORP1L KO, and ORP1L/1S DKO cells. TAG, triacylglycerol. **f** Quantitation of cholesteryl [14C]-esters formation in **e** by densitometry. **g** Western blotting analysis of HeLa, ORP1L KO, and ORP1L/ORP1S DKO cells starved in EBSS and then chased with FBS. **h** Densitometry of p-S6K/S6K in **g**. All values are means ± s.d. (error bar) (*$p < 0.05$; **$p < 0.01$; ***$p < 0.001$; ns, not significant; $n = 3$–10) Statistical analyses use one-way ANOVA. **i** qRT-PCR analysis of PI3KC2β knockdown in ORP1L KO and HeLa control cells. **j** Cholesterol esterification assay in ORP1L KO and HeLa control cells with or without siPI3KC2β knockdown. **k** Quantitation of cholesteryl [14C]-esters formation in **j** by densitometry

esters. ORP1L-null, and especially ORP1L/1S double knockout (DKO) cells displayed a significant reduction in cholesterol esterification compared to control cells, indicating reduced cholesterol availability for acyl-CoA cholesterol acyltransferase (ACAT) (Fig. 2e, f). We also re-expressed ORP1S, ORP1L or both ORP1S and ORP1L in ORP1 DKO cells and carried out lysosomal cholesterol release assay (Supplementary Fig. 4e, f). It appeared that the expression of either ORP1S or ORP1L could facilitate LEL cholesterol exit in the DKO cells, while a more robust effect was observed in the cells re-expressing both ORP1S and ORP1L. Last, a recent study suggested that LEL cholesterol can activate mTORC1 signaling[6]. Indeed, mTORC1 signaling was activated in

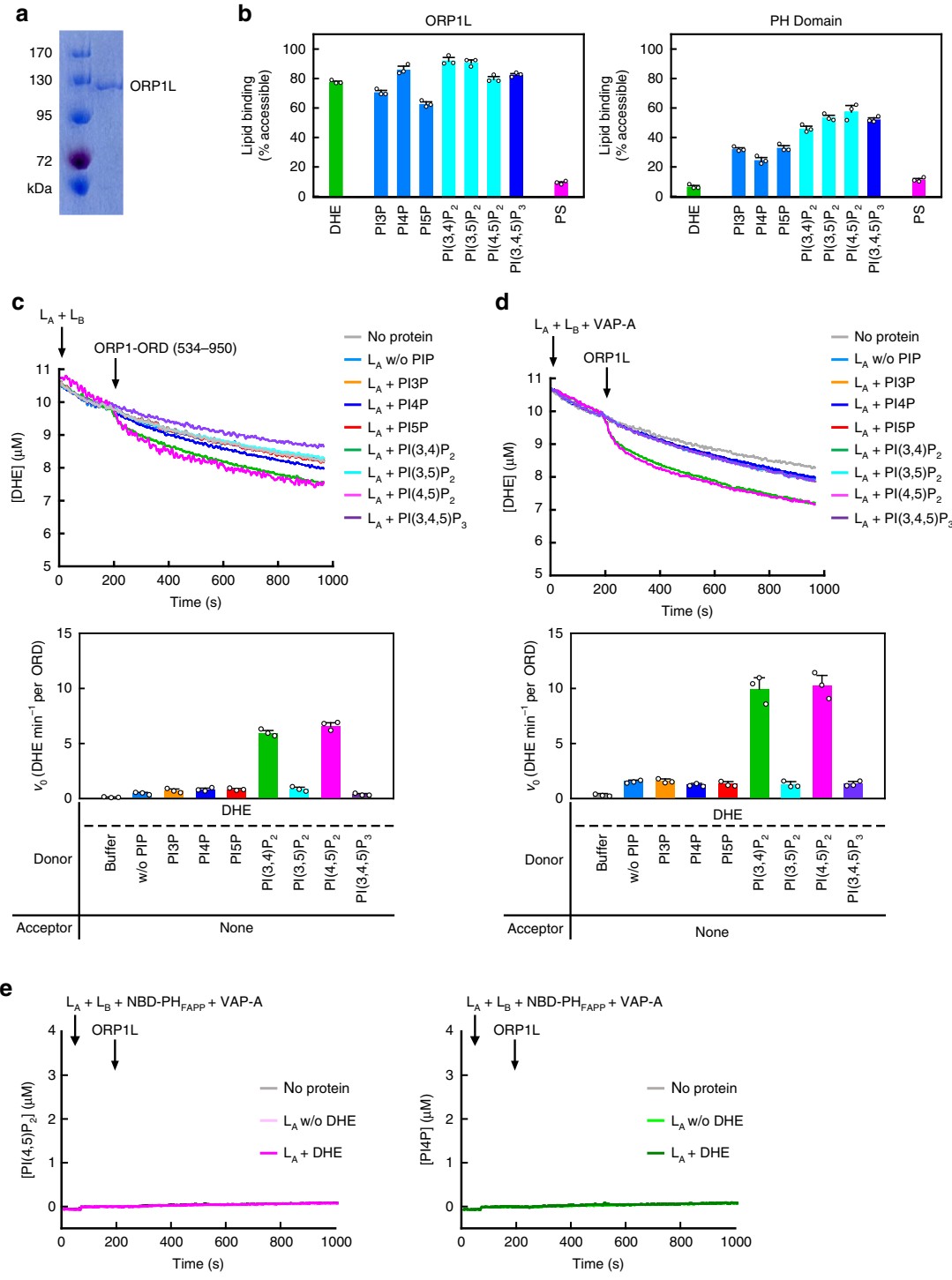

**Fig. 3** PI(3,4)P$_2$/PI(4,5)P$_2$ stimulates cholesterol transport by ORP1L in vitro. **a** SDS-PAGE of full-length ORP1L. **b** Lipid-binding analyses for ORP1L and its PH Domain. **c**, **d** DHE transport assays for ORP1-ORD (**c**) and ORP1L (**d**) with PIP embedded in the donor liposomes. Shown at the bottom are the initial velocities. **e** PI(4,5)P$_2$ (left) and PI4P (right) transport assays for ORP1L. (For all bar graphs, data are shown as mean ± s.e.m. (error bar), $n = 3$)

ORP1L and ORP1L/S null cells (Fig. 2g, h). Taken together, these data demonstrate that ORP1L contributes to the removal of LELs cholesterol, and regulates mTORC1 signaling.

A recent report identified a class II PI3-kinase β (PI3KC2β) as a key enzyme in the synthesis of PI(3,4)P$_2$ on lysosomes[5]. Since our data in vitro demonstrated that PI(3,4)P$_2$ can regulate cholesterol transport through ORP1, we examined the role of this enzyme in endosomal cholesterol trafficking. Knocking down PI3KC2β from HeLa cells dramatically decreased the level of PI

(3,4)P$_2$, which led to a delay of cholesterol efflux from LELs in the U18666A-based cholesterol efflux assay as in ORP1L deficient cells (Fig. 2i and Supplementary Fig. 4g, h). Consistent with the microscopy results, ACAT activity was moderately but significantly reduced in PI3KC2β knockdown cells (Fig. 2j, k). Interestingly, there was no further reduction of cholesterol esterification in ORP1L deficient cells upon PI3KC2β knockdown. These results suggest that PI3KC2β and ORP1L may function in the same pathway in regulating endosomal cholesterol

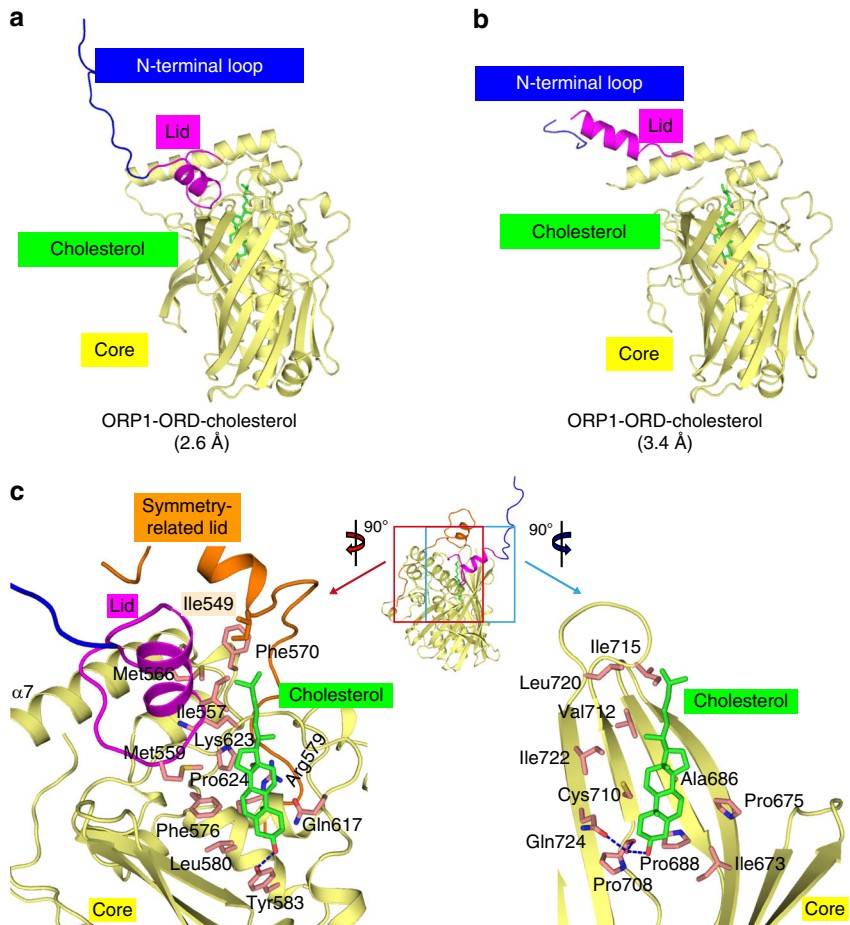

**Fig. 4** Structures of cholesterol-bound ORP1-ORD in open conformations. **a**, **b** Overall structures of ORP1-ORD in complex with cholesterol at 2.6 Å (**a**) and at 3.4 Å (**b**). The cholesterol ligand is highlighted as green stick. **c** Detailed binding mode of cholesterol in the 2.6 Å structure. For clarity, part of the structure has been removed to show the ligand-binding tunnel. Dashed line, polar interaction

efflux. Together, these results suggest that while ORP1L is clearly required in removing LEL cholesterol, its activity may be subjected to the regulation by lysosomal PI(3,4)P$_2$.

**ORP1L efficiently transfers cholesterol in vitro.** The full-length ORP1L contains additional functional domains, which might accelerate the transfer of cholesterol by ORP1. We thus purified the full-length ORP1L protein using HEK293T cells (Fig. 3a), and carried out the lipid-binding assays as in Fig. 1b. Similar to the ORD alone, ORP1L binds DHE/cholesterol and various PIPs; however, ORP1L binds PS weakly, which can be attributed to its PH domain (Figs. 1b and 3b). Considering that ORP1L and PI3KC2β appear to regulate cholesterol homeostasis at the same membrane compartment, we hypothesize that PIPs may regulate the transport of cholesterol on the same membrane. Indeed, the ORP1-ORD-mediated DHE transport were significantly enhanced when the donor L$_A$ liposomes were supplemented with PI(3,4)P$_2$ or PI(4,5)P$_2$, while other PIPs had little effect (Fig. 3c). In the modified assay for ORP1L, the His-tagged VAP-A protein, a FFAT-interacting fragment, was included, which can be recruited to the acceptor liposomes doped with DOGS-NiNTA. As expected, PI(3,4)P$_2$ and PI(4,5)P$_2$ could significantly enhance the basal DHE transport by ORP1L (Fig. 3d). The FFAT motif of ORP1L can interact with VAP-A, which is important for the formation of membrane contact site[26,27]. Consistently, the initial velocity of basal sterol transfer (w/o PIP) by the full-length ORP1L is 3~4-fold higher than that by the ORD alone (Fig. 3c, d).

On the other hand, ORP1L cannot transfer PI4P nor PI(4,5)P$_2$ in the absence or presence of DHE/cholesterol, the same as the ORD alone (Figs. 1d and 3e). Thus, PI(3,4)P$_2$/PI(4,5)P$_2$ on either the donor or acceptor membrane can stimulate cholesterol transport by ORP1. The in vitro DHE transport assays hereafter were performed with PI(3,4)P$_2$ supplemented on the donor liposomes.

**Structures of cholesterol-bound ORP1-ORD.** To further illustrate the mechanisms underlying cholesterol transport by ORP1-ORD, we solved two crystal structures of ORP1-ORD (524–950 or 534–950) in complex with cholesterol (Fig. 4a, b and Table 1). The most striking feature is that their N-terminal loops (residues 514–546) and lid regions (547–565) adopt different open conformations, distinct from the closed conformations observed in yeast Osh structures (Supplementary Fig. 5a)[15,23,25,28,29]. To accomplish the lipid transport function, the ORP/Osh proteins were expected to adopt the open state when extracting lipids from or unloading them onto the membrane, and the closed state when transporting lipids between two membranes[28]. Therefore, the open conformations of ORP1-ORD may represent certain important states when ORP1 exerts physiological functions (see below). On the other hand, the structural core of ORP1-ORD (residues 566–950) is highly conserved from yeast to human.

The 2Fo-Fc map revealed continuous electron density for cholesterol, which binds deeply into the central tunnel in the core (Supplementary Fig. 5b). In the 2.6 Å structure, the 3-hydroxyl group of cholesterol forms direct or water-mediated hydrogen

**Table 1 Data collection and refinement statistics (molecular replacement)**

|  | ORP1-ORD-cholesterol (2.6 Å) | ORP1-ORD-cholesterol (3.4 Å) | ORP1-ORD-PI(4,5)P₂ |
|---|---|---|---|
| *Data collection*[a] |  |  |  |
| Space group | $P3_121$ | $P4_132$ | $P4_32_12$ |
| Cell dimensions |  |  |  |
|     *a, b, c* (Å) | 135.8, 135.8, 91.2 | 149.2, 149.2, 149.2 | 188.7, 188.7, 64.6 |
|     *α, β, γ* (°) | 90, 90, 120 | 90, 90, 90 | 90, 90, 90 |
| Resolution (Å) | 50.00–2.60 (2.64–2.60)[b] | 50.00–3.40 (3.52–3.40) | 50.00–2.70 (2.75–2.70) |
| $R_{sym}$ or $R_{merge}$ | 0.124 (0.617) | 0.074 (0.549) | 0.094 (0.621) |
| $I/\sigma I$ | 24.2 (4.9) | 41.7 (8.1) | 37.4 (2.8) |
| Completeness (%) | 100.0 (100.0) | 99.8 (100.0) | 96.4 (89.0) |
| Redundancy | 12.1 (12.3) | 17.9 (18.1) | 11.7 (4.8) |
| *Refinement* |  |  |  |
| Resolution (Å) | 37.84–2.60 (2.69–2.60) | 34.24–3.40 (3.52–3.40) | 32.58–2.70 (2.78–2.70) |
| No. reflections | 30261 (1478) | 8299 (793) | 32031 (1451) |
| $R_{work}/R_{free}$ | 0.178/0.200 | 0.272/0.307 | 0.245/0.266 |
| No. atoms |  |  |  |
|     Protein | 3225 | 2918 | 6223 |
|     Ligand/ion | 73 | 28 | 131 |
|     Water | 129 | – | 28 |
| *B*-factors (Å²) |  |  |  |
|     Protein | 44.3 | 110.2 | 60.7 |
|     Ligand/ion | 50.8 | 107.2 | 86.8 |
|     Water | 42.0 | – | 34.1 |
| R.m.s. deviations |  |  |  |
|     Bond lengths (Å) | 0.008 | 0.002 | 0.008 |
|     Bond angles (°) | 0.970 | 0.440 | 1.010 |

[a]The data set for each structure was collected from a single crystal
[b]Values in parentheses are for highest-resolution shell

bonds with Tyr583 and Gln724 at the bottom of the binding tunnel (Fig. 4c). The rigid, fused four-ring backbone and the hydrocarbon tail tightly bind to multiple hydrophobic residues at the tunnel wall. The hydrocarbon tail of cholesterol also forms hydrophobic contacts with Ile557 and Met559 in the lid region, as well as Ile549 from the lid of a symmetry-related molecule (Fig. 4c, left panel). In the 3.4 Å structure, the sterol molecule is similarly accommodated in the hydrophobic tunnel (Supplementary Fig. 5c). Moreover, the ORD domains of ORP1, Osh1p and Osh4p, in spite of the poor sequence conservation, display similar spatial distribution of hydrophobic interacting residues in the binding tunnel, which leads to the conserved head-down binding mode for sterols (Supplementary Figs. 1 and 5b). Notably, however, the orientation of cholesterol bound to ORP1 and Osh1p is different from that of various sterols bound to Osh4p in that the tetracyclic rings flip ~180° along the long axis. Thus, the binding tunnels in the ORP/Osh ORD domains are capable of accommodating the sterol ligands in the conserved head-down mode, yet in different orientations along the long axis.

**Three oligomerization states of ORP1-ORD.** To prepare protein–lipid complexes for crystallization trial, ORP1-ORD was mixed directly with ligands or incubated with liposomes doped with indicated lipids. In the gel filtration analyses, the ligand-free ORP1S exists in three different states: monomers, dimers and trimers (Fig. 5a). Binding of cholesterol leads to a shift towards monomer, regardless of the complex preparation procedures. This homo-oligomerization property was barely affected by

truncation of the N-terminal 20 amino acids of ORP1S (Supplementary Fig. 6a). We further examined the oligomeric states of ORP1-ORD in vivo (Fig. 5b). In the presence of crosslinker, GFP-ORP1S (~70 kDa) appeared to form oligomers with the sizes ranging from 150 to 220 kDa. These results suggest that the monomeric, dimeric and trimeric forms all represent physiological oligomeric states of ORP1.

Consistently, structural analyses of the open cholesterol-bound ORP1-ORD revealed domain swapping between symmetry-related molecules, effectively generating dimers or trimers (Fig. 5c, d). In these ORP1-ORD oligomers, the N-terminal loop of one molecule is nestled in a shallow groove on the core of a symmetry-related molecule, forming multiple hydrophobic and polar interactions (Fig. 5e). Such an interface is also observed in the structures of monomeric Osh1p and Osh3p that contain the extended N-terminal loops in their ORD domains[25,29]. We thus constructed a model of ORP1-ORD monomer based on the 2.6 Å cholesterol-bound structure and performed molecular dynamics (MD) simulation (Supplementary Movie 1). During the 100 ns simulation, the lid exhibits the most prominent motion and high flexibility, whereas conformations of the N-terminal loop and the core barely change (Fig. 5f and Supplementary Fig. 6b). Significantly, the lid moves toward the tunnel entrance within 20 ns, and the monomeric ORP1-ORD conformations throughout the remaining simulation are similar to the crystal structures of closed Osh proteins. Therefore, three oligomerization states of the cholesterol-bound ORP1-ORD in solution were also observed in crystals and MD simulations.

Interestingly, the cholesterol molecule rotates in the hydrophobic tunnel of ORP1-ORD by more than 100° during simulation (Supplementary Fig. 6c). We quantified the rotation of cholesterol by measuring the distance between two atoms throughout the simulation (Supplementary Fig. 6d). One atom is the 21st carbon of cholesterol whose position is changed dramatically during MD, and the other is the Cα of Arg579 on the core of ORP1-ORD which is very stable during the simulation. Before the lid closed, the bound cholesterol reaches a steady state (within 5 ns). Thus, it seems that cholesterol in our structure adopts an intermediate state.

The hydrophobic tunnel of ORP/Osh is expected to be sealed when transporting lipids in cytosol. In the closed MD model of ORP1-ORD, several hydrophobic residues from the lid interact with the β4–β5 and β6–β7 loops on the rim of the tunnel, and the closed lid also forms contacts with the hydrophobic cholesterol tail (Supplementary Fig. 6e). In the symmetric cholesterol-bound homodimer, two lid regions, particularly the α1 helices, interact with each other through an extensive hydrophobic interface and completely obstruct the binding tunnels in both ORD cores (Supplementary Fig. 6f). In the homotrimer, the ligand uptake/release path of one molecule is spatially hindered by a symmetry-related lid (Supplementary Fig. 6g). Thus, the lids in three oligomerization states of the cholesterol-bound ORP1-ORD, though adopting different conformations, can seal the lipid-binding tunnel. The domain-swapping mechanism of ORP1 might play a role in transporting cholesterol between membranes.

**Mutational analysis of cholesterol binding site.** To confirm the structural observations, we generated a series of point mutations on ORP1-ORD; however, substitution of many key residues in the hydrophobic lipid-binding tunnel resulted in completely insoluble or partially soluble recombinant proteins. Mutants Y583A and P688A, engineered to disrupt interactions with cholesterol at the tunnel bottom, showed marked decrease in DHE binding (Fig. 6a). Mutations of residues on the tunnel wall also exhibited impaired DHE-binding ability. Deletion of the N-terminal loop

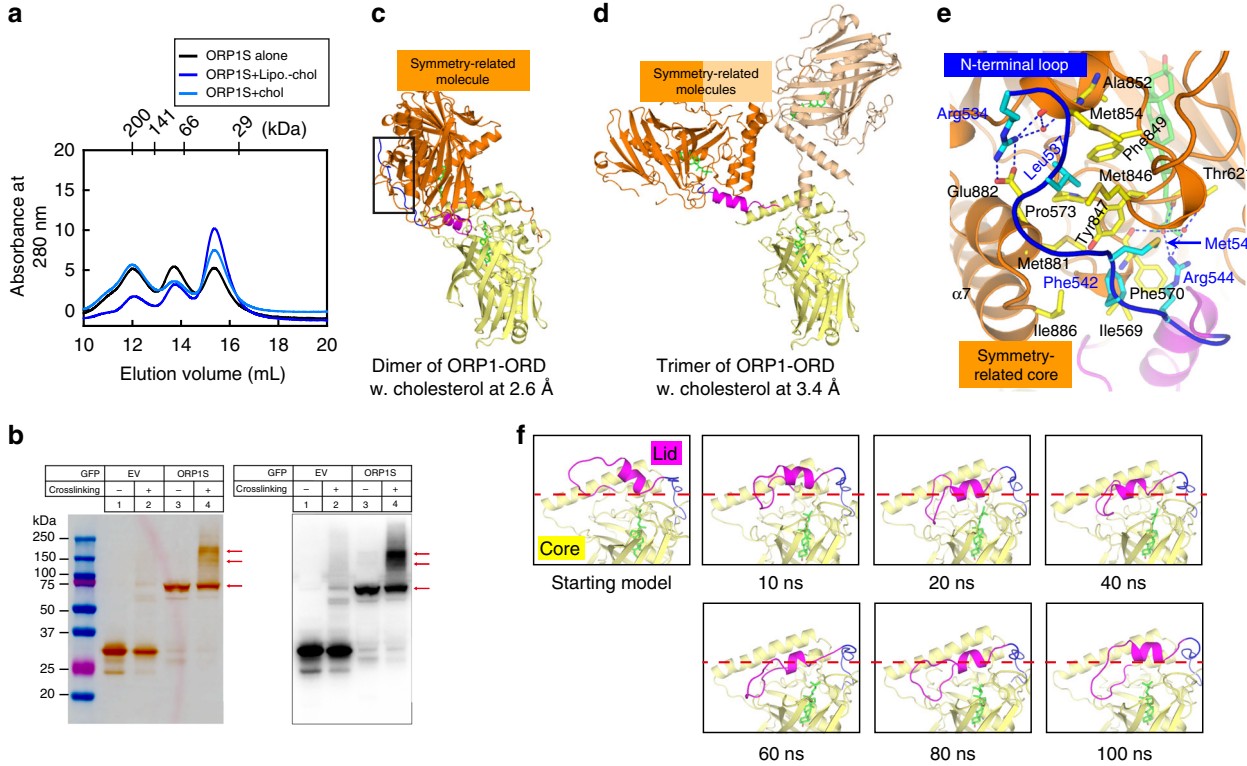

**Fig. 5** ORP1-ORD has three oligomeric states in vitro and in vivo. **a** Gel filtration analyses of ORP1S in isolation and in complex with cholesterol. **b** Immunoblot analysis of the oligomeric states of ORP1S. **c, d** The domain-swapping dimer (**c**) and trimer (**d**) of cholesterol-bound ORP1-ORD. **e** Close-up view of the domain-swapping interactions between the N-terminal loop (blue) and the symmetry-related core (orange) in the dimer shown in **c**. **f** Snapshots at indicated time points during the MD simulations for monomeric ORP1-ORD complexed with cholesterol

and substitution of the interacting residues also dramatically reduced the solubility of ORP1-ORD mutant proteins, yet further deletion of the lid gave rise to a highly soluble fragment. We also engineered point mutations of the lid to interfere with tunnel sealing and/or cholesterol binding. The core alone almost completely lost its ability of DHE binding, and the lid mutations also led to marked or modest decrease of DHE binding (Fig. 6a). Unexpectedly, these mutations had little effects on the binding of $PI(4,5)P_2$, indicating that cholesterol and $PI(3,4)P_2/PI(4,5)P_2$ may bind to ORP1-ORD via relatively independent sites (Supplementary Fig. 7, and see below). We also assessed the effects of these mutants on the $PI(3,4)P_2$-stimulated DHE transport (Fig. 6b). Substitution of residues in the tunnel, particularly those at the bottom, led to sharp or modest reduction in DHE transport. The core alone almost completely lost its ability to transport sterols, and mutations of the lid region also displayed markedly impaired DHE transport abilities (Fig. 6b). Together, these results indicate that the N-terminal loop, the lid and the cholesterol binding tunnel are all indispensable for ORP1-ORD function as a cholesterol transporter.

**Basic surface patches for membrane association.** In addition to ligand binding, the lipid transport/sensing by ORP/Osh proteins requires their association with biological membranes. In the liposome association assay, ORP1-ORD could not bind liposomes containing only DOPC, whereas ~50% ORP1-ORD proteins were associated with liposomes containing 30% PS that mimic the biological membranes enriched in negatively-charged lipids (Fig. 7a). This data clearly demonstrated that ORP1-ORD binds to liposome/membrane mainly through electrostatic interactions. The lid has been suggested to serve as a membrane-binding motif[30], and three positively charged residues within the lid

region are indeed exposed in both the open and closed conformations (Fig. 7b and Supplementary Fig. 8a). There are additional basic surface patches surrounding the tunnel entrance: the C-terminus of helix α7 (Patch I) and the β4–β5 hairpin (Patch II). Moreover, it has been reported that the yeast Osh4p has a distal membrane-binding site on the opposite side of the core[31], and we also found two basic loops on ORP1-ORD distal surface: the β9–β10 loop (Patch III) and the β11–β12 loop (Patch IV, largely disordered in our structures) (Fig. 7b and Supplementary Fig. 8a). Thus ORP1-ORD contains five potential membrane-binding sites.

We mutated key residues in these basic patches, and all soluble mutants were subjected to the liposome association assay (Fig. 7c and Supplementary Fig. 8b). The core alone significantly reduced the membrane-bound fraction, but substitution of basic residues in the lid only resulted in slight decrease. More importantly, membrane association of ORP1-ORD proteins were decreased when those solvent-exposed positive side chains in Patches I and II were mutated. These mutants were also subjected to $PI(3,4)P_2$-stimulated DHE transport assay (Fig. 7d). Consistent with the membrane targeting data, disruption of Patches III and IV had little effect on cholesterol transport. Therefore, the basic Patches I and II in close proximity to the tunnel entrance, as well as the lid region, are essential for lipid transfer by ORP1-ORD through membrane targeting.

**$PI(3,4)P_2$ enhances membrane targeting and sterol extraction.** Considering that ORP1-ORD cannot transfer PIPs yet $PI(4,5)P_2$ or $PI(3,4)P_2$ can stimulate cholesterol/DHE transport, we propose that $PI(4,5)P_2/PI(3,4)P_2$ might facilitate the membrane targeting of ORP1-ORD. The membrane association assay was modified to use DOPC/PS liposomes supplemented with 0.5% indicated lipids

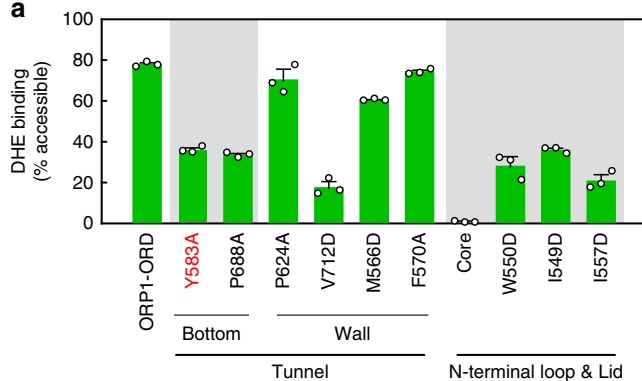

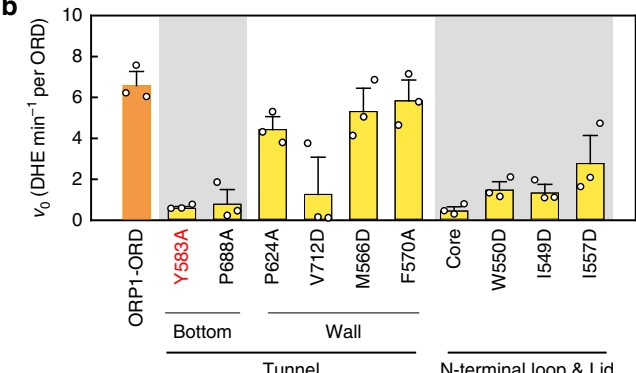

**Fig. 6** Mutational analysis of the cholesterol binding site on ORP1-ORD. **a** DHE-binding assays of ORP1-ORD mutants. **b** PI(3,4)P2-stimulated DHE transport assays of ORP1-ORD mutations with both lipids embedded in donor liposomes. (Data are shown as mean ± s.e.m. (error bar), $n = 3$)

where the ratio of ORP/Osh protein and indicated lipid is 1:5 (Fig. 8a and Supplementary Fig. 9a, b). Strikingly, liposomes supplemented with PI(3,4)P2 recruited ~90% proteins, clearly indicating that ORP1-ORD binds more tightly to the PI(3,4)P2-doped liposome. Unexpectedly, in the presence of DHE and PI4P, the liposome-bound ORP1-ORD dramatically decreased, which is likely due to the extraction of DHE or PI4P from liposomes by ORP1-ORD, similar to Osh4p (Supplementary Fig. 9c). These data suggest that ORP1-ORD, instead of extracting PI(3,4)P2, binds tightly to PI(3,4)P2 embedded in liposomes.

Interestingly, in the modified membrane association assay using liposomes doped with both DHE and PI(3,4)P2, ORP1-ORD could extract DHE and subsequently dissociate from liposomes (Fig. 8a, last two lanes and Supplementary Fig. 9b). Thus, though ORP1-ORD binds tightly to PI(3,4)P2 embedded in liposomes, the presence of DHE/cholesterol can loosen the grip of ORP1-ORD on PI(3,4)P2. In this case, we could perform kinetic analyses to assess the impact of PI(3,4)P2 on membrane binding and cholesterol extraction by ORP1-ORD (Fig. 8b). The initial velocity equations can be derived according to the scheme involving three key steps: (1) association of apo-ORP1 with liposome, (2) transfer of DHE from liposome to ORP1, and (3) dissociation of the DHE-bound ORP1 from liposomes (Supplementary Fig. 9d, e and see the Methods section for details). In the absence of PI(3,4)P2, the initial velocity of DHE binding showed hyperbolic dependence on the ORP1-ORD concentration, and the dissociation constant between protein and liposome ($K_1$) and the DHE extraction rate ($k_{+2}$) can be obtained by fitting the experimental data with Eq. (5). When the DHE-containing liposomes were supplemented with PI(3,4)P2, the data were fit with Eq. (7). The $K_1$ decreased and $k_{+2}$ increased, resulting in

3~4-fold increase in the second order rate constant, $k_{+2}/K_1$. However, the final DHE extraction amounts with or without PI(3,4)P2 were almost the same (Fig. 8c). These results demonstrated that PI(3,4)P2 can improve both the binding affinity between ORP1-ORD and membrane and the extraction rate of cholesterol from membrane to ORP1-ORD, but had little effect on the amount of DHE extraction. Therefore, PI(3,4)P2 can allosterically stimulate sterol extraction by ORP1.

**Role of PI(3,4)P2/PI(4,5)P2 headgroup**. To understand the mechanism underlying membrane association by ORP1-ORD, we tried to determine the structure of ORP1-ORD bound to the PI(3,4)P2/PI(4,5)P2-containing liposomes, but all efforts were unsuccessful. Nevertheless, we solved the PI(4,5)P2-bound structure using the direct mixture of ORP1-ORD and brain PI(4,5)P2 (Fig. 8d, Table 1 and Supplementary Fig. 10a, b). The binding mode of PI(4,5)P2 acyl chains is distinct from those observed in the PI4P-bound Osh structures (Supplementary Fig. 10c, d). Nevertheless, the PI headgroup of PI(4,5)P2 nestles in a polar surface at the tunnel entrance, similar to that in the PI4P-bound Osh structures (Fig. 8e). The 4-phosphate group of PI(4,5)P2 interacts with the highly conserved His651 and His652 in the β2–β3 loop and Arg901 from helix α7, and the additional 5-phosphate group further interacts with His652 and the 4-phosphate group. The inositol ring and 1-phosphate group are engaged in contacts with several charged or polar side chains from the α4–β1 loop and α7 helix. Intriguingly, the inositol ring of PI(4,5)P2 is accommodated in ORP1-ORD with ~180° rotation compared with that of PI4P in Osh proteins; otherwise the 5-phosphate group would have caused a steric clash with helix α7 (Supplementary Fig. 10c). The rotatable property of the PI headgroup provides the possibility that ORP1-ORD can bind other PIPs such as PI(3,4)P2 that stimulates cholesterol transport as well (Fig. 1). More significantly, the presence of such a charged, protruding group (5-phosphate) should interfere with the closure of the lid, which is likely to be very important for ORP1 functioning as a PI(4,5)P2- or PI(3,4)P2-dependent cholesterol transporter, similar to the case of another LTP[32].

Considering that the solvent-exposed moiety of membrane-bound PI(3,4)P2/PI(4,5)P2 is the negatively charged PI headgroup, we first replaced the polar residues at tunnel entrance and performed the membrane association assay using PI(3,4)P2-containing liposomes (Fig. 8f and Supplementary Fig. 8b). Indeed, substitution of residues involved in PI headgroup recognition severely affected membrane targeting of ORP1-ORD. By contrast, these mutations barely interfered with DHE/cholesterol binding, again suggesting that PIPs (particularly the PI headgroup) and cholesterol bind to separate sites of ORP1-ORD (Supplementary Fig. 11a). Interestingly, the bound fractions of these polar mutants (~50%) were almost the same as that of the wild-type ORP1-ORD bound to the DOPC/PS liposomes (Figs. 7c and 8f). Moreover, mutants of hydrophobic residues at the rim of tunnel entrance, as well as charged residues in the lid and basic patches, displayed similar binding ability to the PI(3,4)P2-containing liposomes as wild-type ORP1-ORD (Fig. 8f and Supplementary Fig. 11b). These data demonstrated that the lid and basic Patches I, II are required for ORP1-ORD targeting to negatively charged membranes, whereas the polar interface for PI headgroup binding is required for recruiting ORP1-ORD specifically to PI(4,5)P2/PI(3,4)P2-containing sites. Consistent with the binding results, the initial rate of PI(3,4)P2-stimulated DHE transport also significantly decreased when the polar residues at the tunnel entrance, but not the hydrophobic side chains, were mutated (Fig. 8g). Together, our biochemical and structural results showed that PI(3,4)P2 binds to ORP1-ORD independently of cholesterol and can

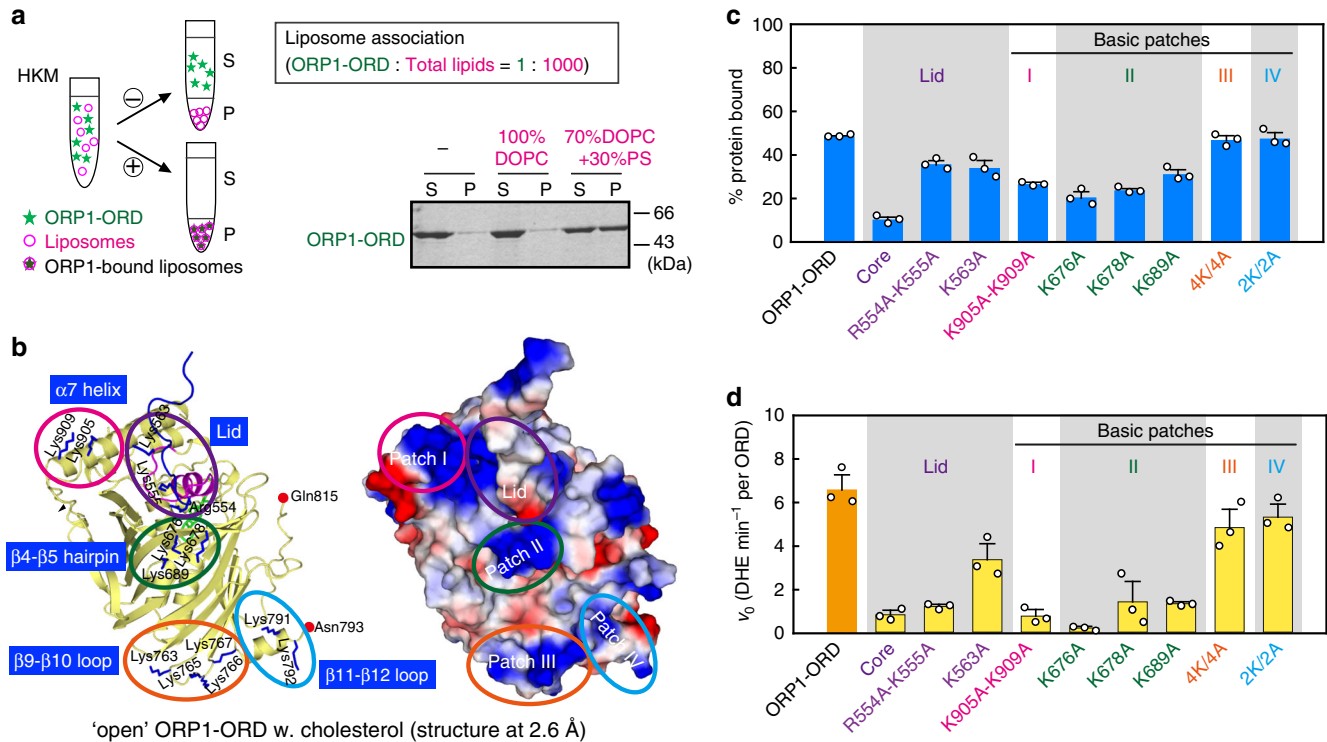

**Fig. 7** Key regions for membrane association of ORP1-ORD. **a** Liposome association assay for ORP1-ORD. **b** Potential membrane-binding sites on the open ORP1-ORD structure. The surface representation is colored according to electrostatic potential (positive, blue; negative, red), and the lid region and four basic patches are circled. **c** Bar graph for membrane-bound fractions of basic region mutants to liposome enriched in negatively charged lipids. **d** Effects of basic region mutations on the initial rates of PI(3,4)P$_2$-dependent DHE transport. (For all bar graphs, data are shown as mean ± s.e.m. (error bar), $n = 3$)

stimulate the transport of cholesterol, demonstrating that PI(3,4)P$_2$ acts as an allosteric activator of ORP1-mediated cholesterol transport.

**Working model for PI(3,4)P$_2$-stimulated cholesterol transport.** To examine importance of the aforementioned sites for ORP1-ORD function in vivo, we transfected ORP1L KO cells with key ORP1L mutants and performed U18666A-mediated lysosomal cholesterol release assay as described in Fig. 2c. Compared to empty-vector control cells, those cells overexpressing wild-type ORP1L displayed much lower level of accumulated LEL cholesterol induced by U18666A, indicating that ORP1L is able to facilitate efficient removal of LEL cholesterol (Fig. 9a, b). However, this function of ORP1L appeared to be markedly impaired by mutations designed to interfere with cholesterol binding and/or membrane association of ORP1-ORD. These in vivo data further highlight that the intact ORP1-ORD function requires the hydrophobic tunnel for cholesterol binding, the basic surface patches for membrane association, and more importantly, the polar interface at tunnel entrance for specific targeting to PI(4,5)P$_2$/PI(3,4)P$_2$-containing membrane.

Based on results presented in this study and previous knowledge on ORP/Osh family, we propose a working model for the PI(4,5)P$_2$/PI(3,4)P$_2$-enhanced cholesterol transport by ORP1-ORD which involves multiple conformational changes (Fig. 9c). ORP1-ORD can be recruited to membrane by electrostatic interactions between negatively charged lipids and the basic Patches I, II and lid region of ORP1-ORD, and thus slowly extracts (or unloads) cholesterol (basal). PI(4,5)P$_2$ or PI(3,4)P$_2$ could specifically target ORP1 to certain membrane domains through massive interactions between the PI(4,5)P$_2$/PI(3,4)P$_2$ headgroup and the polar interface at ORP1-ORD tunnel

entrance, and orientate ORP1-ORD in the right position with the lipid-binding tunnel of ORP1-ORD facing the membrane. The presence of a charged, protruding group (the 3- or 5-phosphate of PI(3,4)P$_2$ or PI(4,5)P$_2$) would also facilitate cholesterol extraction/unloading by pushing the lid open. ORP1-ORD then dissociates from the membrane. In cytosol, either in the apo/unbound or the cholesterol-bound form, ORP1-ORD can adopt the closed, monomeric conformation, as well as the open conformations by forming domain-swapped homodimer or homotrimer.

## Discussion

OSBP and ORPs/Oshs are major lipid-binding proteins in eukaryotic cells that mediate the signaling and/or intermembrane transport of lipids. Whether ORP1L functions as a cholesterol sensor or transporter remains controversial[3,4,20]. Here, while we cannot rule out its cholesterol sensor function, our structural, biochemical and cell biological data suggest that ORP1L is a bona fide cholesterol transporter. However, cholesterol exit from LELs was only hampered slightly in ORP1L deficient cells, suggesting the presence of ORP1L-independent mechanisms. For instance, lysosomes and peroxisomes may form contacts to facilitate lysosomal cholesterol transfer to the ER[33]. ORP5 was also implicated in LEL to ER cholesterol transport[34]. The transport function of ORP/Osh proteins has been elegantly demonstrated to involve PI4P: the forward transport of cargo lipids from the ER (site of synthesis) to the target membrane is coupled with the reciprocal transport of PI4P from the target membrane back to the ER, where it is hydrolyzed by the ER phosphatase Sac1[15,18,23]. The hydrolysis of PI4P is believed to permit the continuous transport of the main lipid cargo against a concentration gradient. Here, our data point to a distinct mechanism

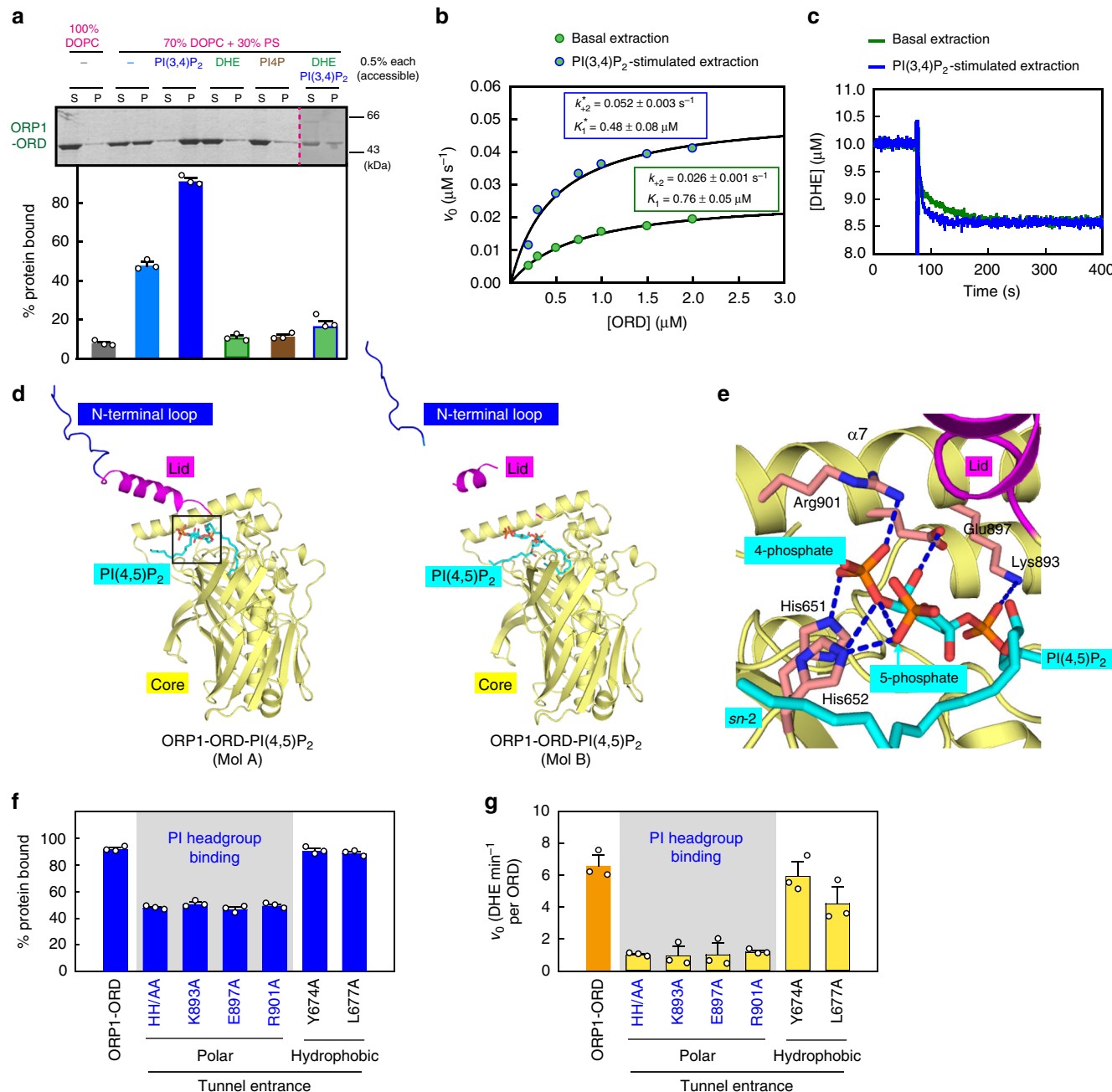

**Fig. 8** PI(3,4)P$_2$/PI(4,5)P$_2$ headgroup facilitates specific membrane targeting. **a** Membrane association analyses with liposomes supplemented with indicated lipids. The ratio of ORP-ORD and indicated lipid is 1:5. The last two lanes are from a separate experiment. **b** Kinetic analyses of DHE extraction in the absence or presence of PI(3,4)P$_2$ (mean ± s.e.m., $n = 3$). **c** Representative time traces of DHE extraction by ORP1-ORD (1.5 μM). **d** Structure of ORP1-ORD in complex with PI(4,5)P$_2$. The PI(4,5)P$_2$ ligand is highlighted as cyan stick. **e** Close-up view of PI(4,5)P$_2$ headgroup binding at the tunnel entrance of ORP1-ORD (Mol A). **f, g** Effects of tunnel entrance mutations on the bound fraction of ORP1-ORD with PI(3,4)P$_2$-containing liposomes (**f**) and the initial rates of PI(3,4)P$_2$-dependent DHE transport with both lipids embedded in the donor liposomes (**g**). (For all bar graphs, data are shown as mean ± s.e.m. (error bar), $n = 3$.)

by which PI(3,4)P$_2$/PI(4,5)P$_2$ may allosterically regulate ORP1-ORD-mediated cholesterol transfer. First, ORP1-ORD does not seem to employ the counter-transport mechanism: while ORP1-ORD is a cholesterol transporter, it cannot transfer any PIP species. Second, PI(3,4)P$_2$/PI(4,5)P$_2$, but not PI4P or other PIPs, can promote specific targeting of ORP1-ORD to certain sites of a membrane. This unique feature may be due to the inability of ORP1-ORD to efficiently extract PI(4,5)P$_2$/PI(3,4)P$_2$ from membranes. Third, PI(3,4)P$_2$/PI(4,5)P$_2$ can also facilitate cholesterol extraction/unloading likely owing to the presence

of 3/5-phosphate of PI(3,4)P$_2$/PI(4,5)P$_2$, which can help maintain an opening state of the lid. Last, cholesterol and PI(3,4)P$_2$/PI(4,5)P$_2$ (the PI headgroup) bind to the lipid-binding tunnel and the tunnel entrance of ORP1-ORD, respectively. In a way, PI(3,4)P$_2$/PI(4,5)P$_2$ can allosterically enhance cholesterol transport by ORP1-ORD.

Interestingly, the in vitro data suggest that PI(3,4)P$_2$/PI(4,5)P$_2$ on either the donor or acceptor membrane can dramatically promote cholesterol transport by ORP1-ORD, and ORP1L indeed regulates lysosomal cholesterol homeostasis with PI(3,4)P$_2$ at

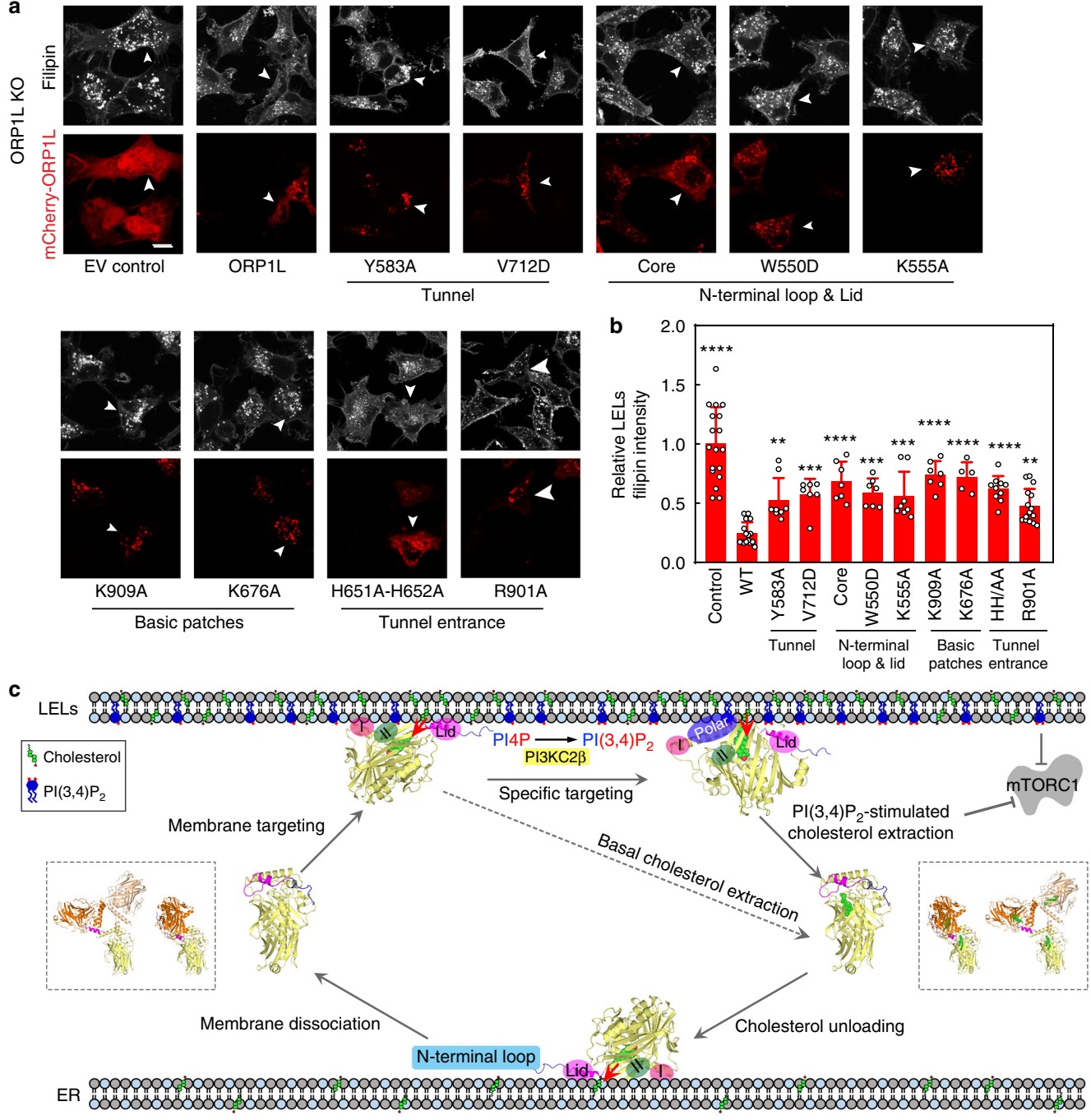

**Fig. 9** PI(3,4)P$_2$/PI(4,5)P$_2$ allosterically stimulates ORP1 function. **a** Lysosomal cholesterol release assay in ORP1L KO cells overexpressing ORP1L mutants (scale bar = 10 μm). **b** Quantitation of LELs filipin intensities of transfected cells in **a** (data are shown as mean ± s.d. (error bar); n = 5–15). **c** Working model for PI(4,5)P$_2$/PI(3,4)P$_2$-stimulated cholesterol transport by ORP1-ORD. The red arrows indicate the orientations of lipid-binding tunnel to membrane

the same limiting membrane of LELs. Importantly, perhaps a counter-transport mechanism is not needed in the case of ORP1L, which moves cholesterol downhill from LELs (high cholesterol) to the ER (low cholesterol). In this connection, a recent study demonstrated that PI(3,4)P$_2$ synthesized locally by PI3KC2β accumulates on LEL limiting membrane to repress mTORC1 activity upon growth factor deprivation[5]. While PI(3,4)P$_2$ can repress mTORC1 activity through 14-3-3 proteins as reported, it may also inhibit mTORC1 by recruiting/activating ORP1L to remove cholesterol, since LEL cholesterol can activate mTORC1 through SCL38A9[6]. Indeed, a moderate increase of LEL cholesterol upon ORP1L depletion hyperactivated mTORC1.

PI4P binding activity of ORP1L has been reported to be required for cholesterol removal from LELs[20]. However, our results showed that ORP1-ORD was unable to transport PI4P, although it could bind and extract PI4P. One plausible explanation is that ORP1 may be incapable of unloading PI4P to membrane. The PIP delivery assays showed that ORP1-ORD cannot deliver PI4P or PI(4,5)P$_2$ to liposome, regardless of the presence of DHE (Supplementary Fig. 12a). Consistent with the cholesterol-PI4P counter-transport mechanism, Osh4p can weakly deliver PI4P and PI(4,5)P$_2$, but only the PI4P delivery can be enhanced by DHE/cholesterol embedded in the acceptor liposome. Therefore, we believe that ORP1 cannot transport PI4P

because the ORP1-bound PI4P cannot be released from ORP1-ORD and unloaded to membrane. Consequently, instead of stimulation, PI4P could competitively inhibit DHE transfer by ORP1 in vitro (Supplementary Fig. 12b). It should also be noted that PI4P on the LEL limiting membrane is of low abundance, by tens and hundreds times lower than the level of cholesterol[35–38]. Thus, PI4P might not hinder cholesterol binding and transport by ORP1 in vivo.

Taken together, our results here unveil a distinct mechanism by which PIPs may allosterically regulate the efficiency of OSBP/ORPs-mediated transport of major lipid species such as cholesterol, but not as a cargo.

## Methods

**Lipids**. Cholesterol (catalog no. 700000P), DOPC (1,2-dioleoyl-sn-glycero-3-phosphocholine) (catalog no. 850375P), DOPE (1,2-dioleoyl-sn-glycero-3-phosphoethanolamine) (catalog no. 850725P), PS (1,2-dioleoyl-snglycero-3-phosphoserine) (catalog no. 840035P), brain PI(4,5)P$_2$ (L-α-phosphatidylinositol-4,5-bisphosphate) (catalog no. 840046P), brain PI4P(L-α-phosphatidylinositol-4-phosphate) (catalog no. 840045P), liver PI (L-α-PI) (catalog no. 840042P), 18:1/18:1 PI3P (1,2-dioleoyl-sn-glycero-3-phospho-(1′-myo-inositol-3′-phosphate)) (catalog no. 850150P), 18:1/18:1 PI5P (1,2-dioleoyl-sn-glycero-3-phospho-(1′-myo-inositol-5′-phosphate)) (catalog no. 850152P), 18:1/18:1 PI(3,4)P$_2$ (1,2-dioleoyl-sn-glycero-3-phospho-(1′-myo-inositol-3′,4′-bisphosphate)) (catalog no. 850153P), 18:1/18:1 PI(3,5)P$_2$ (1,2-dioleoyl-sn-glycero-3-phospho-(1′-myo-inositol-3′,5′-bisphosphate)) (catalog no. 850154P), 18:1/18:1 PI(3,4,5)P$_3$ (1,2-dioleoyl-sn-glycero-3-phospho-(1′-myo-inositol-3′,4′,5′-trisphosphate)) (catalog no. 850156P), DNS-PE (1,2-dioleoyl-sn-glycero-3-phosphoethanolamine-N-(5-dimethylamino-1-naphthalenesulfonyl)) (catalog no. 810330C), Rhod-PE (1,2-dipalmitoyl-sn-glycero-3-phosphoethanolamine-N-(lissamine rhodamine B sulfonyl)) (catalog no. 810158P), DOGS-NiNTA (1,2-dioleoyl-sn-glycero-3-[(N-(5-amino-1-carboxypentyl)iminodiacetic acid)succinyl] (nickel salt), catalog no. 790404) were purchased from Avanti Polar Lipids. DHE (catalog no. E2634) was obtained from Sigma-Aldrich.

**Protein preparation**. The ORP1-ORD proteins (residues 514/524/534–950) and human VAP-A (8-212) were subcloned into pET21b vector with C-terminal His$_6$-tag, and the site-specific mutations were generated by QuickChange kit (Agilent). All primers are included in Supplementary Table 1. The Osh4p, PH$_{FAPP}$ (the PH domain of four-phosphate-adaptor protein 1, residues 1–100, with mutations C37S, C94S, and T13C), C2$_{Lact}$ (residues 270–427, with mutations C270A, C472A, and H352C), ORP1-PH domain (214–362) were cloned into pGEX-2T vector with N-terminal GST tag. All constructs were verified by DNA sequencing. All proteins, over-expressed in E. coli (Rosetta(DE3), Novagen) at 22 °C, were purified over Ni-NTA (Qiagen) or GS4B (GE Healthcare) columns and then by ion exchange and gel filtration chromatography (Source-15Q/15S and Superdex-200 10/300, GE Healthcare) at 4 °C. The GST tag was cleaved by thrombin and removed through GS4B columns. The purified proteins in a buffer containing 10 mM Tris-HCl, pH 8.0, 150 mM NaCl and 2 mM DTT were stored at −80 °C and subjected to in vitro biochemical analyses and crystallization trials.

ORP1L was subcloned into pCAG-strep plasmid with N-terminal tandem twin-strep tag. For expression, HEK293T cells (Sigma-Aldrich) in suspension were grown to a density of 1.5–2.0 × 10$^6$ ml$^{-1}$ in SMM 293-TI medium (Sino Biological Inc.) at 37 °C. The temperature was lowered to 30 °C, and the cells were transfected using PEI (Polyetherimide, Polysciences) according to manufacturer's instruction. After 60 h, the cells were harvested and suspended in a buffer containing 10 mM Tris-HCl, pH 8.0, 150 mM NaCl and 1 mM EDTA (supplemented with 1 mM PMSF, 1 mg/ml aprotinin, 1 mg/ml leupeptin and 1 mg/ml pepstatin), and then lysed with Dounce homogenizer. After centrifugation (16,200 × g, 60 min), the ORP1L protein was purified over StrepTactin superflow high capacity resin (IBA Lifesciences) by gravity flow and then by ion exchange (mono Q 5/50, GE Healthcare) and gel filtration chromatography at 4 °C. The purified proteins in a buffer containing 10 mM Tris-HCl, pH 8.0 and 150 mM NaCl were stored at −80 °C.

For the NBD labeling, the DTT-free PH$_{FAPP}$ or C2$_{Lact}$ was mixed with a 10-fold excess of N,N′-dimethyl-N-(iodoacetyl)-N′-(7-nitrobenz-2-oxa-1, 3-diazol-4-yl) ethylenediamine (IANBD-amide, Molecular Probes). After incubation at 4 °C overnight, the reaction was stopped by adding a 10-fold excess of L-cysteine. The free probe was removed by gel filtration chromatography (Superdex-75 10/300, GE Healthcare). The labeling yield (~100%) was estimated from the ratio of the optical density (OD) of tyrosine and tryptophan at 280 nm ($\varepsilon = 29,450$ M$^{-1}$ cm$^{-1}$ for PH$_{FAPP}$, $\varepsilon = 45,045$ M$^{-1}$ cm$^{-1}$ for C2$_{Lact}$) and NBD at 495 nm ($\varepsilon = 25,000$ M$^{-1}$ cm$^{-1}$).

**Liposomes preparation**. Lipids in chloroform were mixed at the desired molar ratio in vials; the solvent was evaporated using nitrogen gas yielding a thin lipid film on the sides of a round bottom flask. The lipid film was thoroughly dried to remove residual organic solvent by placing the vial on a vacuum pump overnight. The films were hydrated in the HKM buffer (50 mM HEPES, pH 7.2, 120 mM potassium acetate, 1 mM MgCl$_2$) to obtain a suspension of multilamellar liposomes, which was subjected to ten freeze-thaw cycles using liquid nitrogen followed by extrusion through polycarbonate filters of 0.2 μm pore size using a mini-extruder (Avanti Polar Lipids). Liposomes were stored at 4 °C in the dark and used within 4 days (or in 2 days when containing light-sensitive lipids, such as DHE, DNS-PE or Rhod-PE).

**DHE-binding assay**. The sample (500 μL) containing DOPC/DOPE (2:1) liposomes (150 μM total lipids) doped with 2.5% DNS-PE and 2% DHE were incubated at 25 °C in a quartz cuvette. The dansyl signals upon DHE excitation at 310 nm (bandwidth 10 nm) were recorded at 525 nm (bandwidth 10 nm) with a fluorescence spectrophotometer Dual-FL (HORIBA) equipped with a magnetic stirrer in the cuvette holder. The signals before and 5 min after the addition of 3 μM proteins correspond to the $F_{max}$ and $F$ values, respectively. The control signal corresponding to the maximal DHE binding ($F_0$) was determined with 5 mM methyl-β-cyclodextrin. The percentage of DHE binding is given by $100 \times (1 - ((F - F_0)/(F_{max} - F_0)))$.

**PIP and PS binding assays**. The sample (500 μL) containing 150 μM DOPC/DOPE liposomes, doped with 4% indicated PIP or PS (3 μM accessible), were mixed with NBD-PH$_{FAPP}$ or NBD-C2$_{Lact}$ (250 nM) at 25 °C in a small quartz cuvette. The NBD spectra were recorded from 505 to 650 nm (bandwidth 10 nm) upon excitation at 490 nm (bandwidth 10 nm) before and 5 min after the injection of 3 μM proteins. The intensities at 525 nm measured before and after the addition of proteins correspond to $F_{max}$ and $F$, respectively. A control signal ($F_0$) was measured with 250 nM NBD-PH$_{FAPP}$ or NBD-C2$_{Lact}$ in the presence of liposomes without PIPs or PS. The contribution of buffer or liposome alone was subtracted from the NBD signal. The percentage of PIP or PS binding is given by $100 \times (1 - ((F - F_0)/(F_{max} - F_0)))$.

**DHE transport assay**. A suspension (465 μL) of L$_A$ liposomes (DOPC/DOPE, 2:1; total lipids, 200 μM), doped with 2.5% DNS-PE, 5% DHE was incubated at 25 °C under constant stirring in HKM buffer. After 1 min, 25 μL of L$_B$ liposomes (DOPC/DOPE, 2:1; total lipids, 200 μM) containing 4% indicated PIP or not was added. After another 3 min, 10 μL protein was injected to final concentration of 100 nM. The concentration of accessible DHE is 10 μM. DHE transport was followed by measuring the dansyl signal at 525 nm (bandwidth 10 nm) upon DHE excitation at 310 nm (bandwidth 10 nm). The decrease in FRET signal between DHE and DNS-PE on L$_A$ was converted into the amount of DHE (in μM) transferred from liposomes L$_A$ to L$_B$: $10 \times ((F - F_0)/(F_{max} - F_0))$, where $F_{max}$ and $F$ are the signals before and after protein was injected and $F_0$ is the signal measured with L$_A$ liposomes containing 2.5% DNS-PE but devoid of DHE. The DHE transport was also measured when both DHE and indicated PIP were embedded in L$_A$ liposomes and the L$_B$ liposomes contain no PIP.

In the modified assay for ORP1L, the donor liposomes (DOPC/DOPE, 2:1; total lipids, 200 μM), containing 2.5% DNS-PE, 5% DHE and 4% PIPs, were incubated with His-tagged VAP-A (1 μM) at 25 °C in HKM buffer. After 1 min, 25 μL of the acceptor liposomes (DOPC/DOPE, 2:1; total lipids, 200 μM), doped with 2% DOGS-NiNTA, were added. After another 3 min incubation, 10 μL ORP1L protein was added (final concentration of 100 nM) to start the DHE transfer.

**PIP transport assay**. A suspension (465 μL) of L$_B$ liposomes (DOPC/DOPE, 2:1; total lipids, 200 μM) containing 2% Rhod-PE and 4% indicated PIP were incubated with 250 nM NBD-PH$_{FAPP}$ at 25 °C in HKM buffer under constant stirring. The concentration of accessible PIP (in the outer leaflet of liposomes) is 4 μM. When bound to PIPs in the donor L$_B$ liposomes, the fluorescence of NBD-PH$_{FAPP}$ is quenched due to FRET with Rhod-PE. Some 25 μL of L$_A$ liposomes (DOPC/DOPE, 2:1; total lipids, 200 μM) doped with 5% DHE or not were added after 1 min, and 10 μL protein (final concentration of 100 nM) was injected after additional 3 min. Transport of PIP from L$_B$ to L$_A$ was followed by measuring the NBD signal at 525 nm (bandwidth 10 nm) upon excitation at 460 nm (bandwidth 10 nm). To determine the amount of transported PIP, we first measured the NBD signal $F_{eq}$ when PIP is fully equilibrated between liposomes: NBD-PH$_{FAPP}$ (250 nM) was mixed with L$_B$ and L$_A$ liposomes that each contains 2% PIP. The fraction of PIP transported to L$_A$ liposomes, PIP$_A$/PIP$_T$, is directly equal to the fraction of PH$_{FAPP}$ bound on L$_A$ liposomes and corresponds to $F_{Norm} = 0.5 \times ((F - F_0)/(F_{eq} - F_0))$ where $F_0$ is the NBD signal prior to the addition of protein. The amount of PIP (in μM) transferred from liposomes L$_B$ to L$_A$ corresponds to $4 \times F_{Norm}$.

In the modified assay for ORP1L, the donor liposomes (DOPC/DOPE, 2:1; total lipids, 200 μM), containing 2% Rhod-PE and 4% indicated PIP, were incubated with 250 nM NBD-PHFAPP and 1 μM His-tagged VAP-A at 25 °C in HKM buffer. Some 25 μL of acceptor liposomes (DOPC/DOPE, 2:1; total lipids, 200 μM), doped with 2% DOGS-NiNTA and 5% DHE, were added after 1 min, and 10 μL protein (100 nM) was added after additional 3 min.

**Cell culture and transfection**. HeLa cells were purchased from the American Type Culture Collection (ATCC HTB-22; Rockville, MD). Monolayers of cells were maintained in medium supplemented with 10% FBS, 100 units/mL penicillin, and 100 μg/mL streptomycin sulfate in 5% CO$_2$ at 37 °C. DNA transfection was performed using Lipofectamine™ LTX and Plus Reagent (Life Technology)

according to manufacturer's instruction. For each transfection, 1–2 μg/well of plasmid cDNA were used in 6-well plates. siRNA transfection was carried out in cells at ~20% confluence according to standard methods using Lipofectamine™ RNAiMAX transfection reagent (Life Technology). Forty pmoles of duplexes of siRNA were used for transfection of one well of cells grown in 6-well plate.

**CRISPR.** HeLa ORP1L knockout (KO) and ORP1L/1S double knockout (DKO) cells were generated by the CRISPR/Cas9 methods[39]. The sgRNA sequences were designed by analyzing exon sequences close to the start codons of ORP1L (NCBI Reference Sequence: NM_080597.3) or ORP1S (NCBI Reference Sequence: NM_018030.4) at http://crispr.mit.edu. DNA oligo sequence 5′AACTATTAGAGACCATGGCG (sgRNA#1) was chosen to knockout ORP1L, and 5′ACAGCGCCTAACTGAATACA (sgRNA#2) was chosen to knockout both ORP1L and ORP1S. Corresponding DNA oligos were synthesized with inframed BbsI sites flanking the forward and revers oligos. The synthesized DNA oligos were annealed and subcloned into the sgRNA expression vector pSpCas9(BB)-2A-GFP (PX458), which was a gift from Feng Zhang (Addgene plasmid # 48138)[39]. PX458-sgRNA#1 and PX458-sgRNA#2 were verified by sequencing and transfected into wild-type HeLa cells. Forty-eight hours after transfection, fluorescence activated cell sorting of HeLa cells were carried out with a BD Influx Cell Sorter (BD Biosciences, Cytopeia, USA) and a single cell with GFP signal was sorted into each well of 96-well plates. Individual cell clones were expanded and screened for the loss of ORP1L and/or ORP1S by western blotting analysis. Three positive ORP1L KO and ORP1L/1S DKO clones were used in this study.

**Immunostaining of PI(3,4)P₂.** All steps were carried out at room temperature. HA-PI3KC2β was a gift from Dr. Volker Haucke (Leibniz-Forschungsinstitut für Molekulare Pharmakologie, Berlin, Germany)[5]. Cells grown on coverslips in 1 ml medium were fixed by the addition of 1 ml 4% paraformaldehyde in PBS for 15 min. Cells were washed three times with PBS containing 50 mM $NH_4Cl$, followed by permeabilization for 5 min with 20 μM digitonin in PBS. After three rinses with PBS to remove digitonin, cells were blocked for 45 min in PBS containing 5% normal goat serum (NGS) and 50 mM $NH_4Cl$. Primary anti-PI(3,4)P₂ antibody (Echelon Biosciences, catalog no. Z-P034; 1:100) were diluted in PBS containing 5% NGS saponin and applied to cells for 1 h. After three washes with PBS, the incubation with secondary antibody diluted in PBS containing 5% NGS were carried out for 45 min. Cells were then washed three times with PBS and post-fixed in 2% paraformaldehyde in PBS for 10 min at room temperature, followed by three washes with PBS containing 50 mM $NH_4Cl$ and one rinse in distilled water. Cells were mounted in ProLong® Gold antifade reagent (Life Technologies) for confocal microscopy.

**Immunoblot analysis.** Samples were mixed with 2 × laemmli buffer and subjected to 7.5% or 10% SDS-PAGE. After electrophoresis, the proteins were transferred to Hybond-C nitrocellulose filters (GE Healthcare). Incubations with primary antibodies were performed at 4 °C overnight. Primary antibodies used were rabbit polyclonal to ORP1 (Abcam, catalog no. ab131165); S6K (Cell Signaling Technology (CST), catalog no. 9202); p-T389 S6K (CST, catalog no. 9205); GAPDH (CST, catalog no. 2118, clone 14C10); LC3B (CST, catalog no. 2775); HA-Tag (CST, catalog no. 3724); and mouse monoclonal to GFP (Santa Cruz Biotechnology, catalog no. sc9996). Secondary antibodies were peroxidase-conjugated AffiniPure donkey anti-rabbit or donkey anti-mouse IgG (H+L; Jackson ImmunoResearch Laboratories) used at a 1:5000 dilution. The bound antibodies were detected by ECL Western blotting detection reagent (GE Healthcare or Merck Millipore) and visualized with Molecular Imager® ChemiDocTM XRS + (Bio-Rad Laboratories).

**Filipin staining and fluorescence microscopy.** Cells grown on coverslips were fixed with 4% paraformaldehyde for 30 min at room temperature. Cells were stained with freshly prepared ~50 μg/mL of filipin (a fluorescent dye that specifically binds to cellular free cholesterol) in PBS for 1 h at room temperature. Stained cells were imaged using an Olympus FV1200 laser-scanning microscope equipped with four detectors and six laser lines (405, 458, 488, 515, 559, and 635 nm). Laser line 405 nm was used to detect filipin signals and 559 nm was used to detect mCherry fluorescence. The manufacturer's software and Fiji software were used for data acquisition and analysis.

**Cholesterol esterification assay.** Monolayer of cells grown in 60-mm dishes were starved in lipoprotein deficient serum medium overnight. Cells were cultured in FBS medium (20%) for 5 h in the absence or presence of U18666A (2.5 μg/mL). $^{14}C$-oleic acid (1 μCi) conjugated with bovine serum albumin was added directly to the medium and cells were chased for further 2 h. Culture media were then removed and cells were processed for lipid extraction, followed by TLC and phosphorimaging[34]. Briefly, the cells were washed twice with Buffer A (0.15 M NaCl, 0.05 M Tris-HCl, 2 mg/mL BSA, pH 7.4) and once with Buffer B (0.15 M NaCl, 0.05 M Tris-HCl, pH 7.4). The cells were then incubated with 1 mL of hexane-isopropanol (3:2) at room temperature for 30 min. The organic solvent was collected and the cells were incubated with another 1 mL of the same solvent for 15 min at room temperature. The two organic solvent extracts were combined in a 2-mL glass vial. The glass vials were kept in the fume hood until the solvent was

evaporated to dryness. The cells remained in each well were harvested using 1 mL of 0.1 M NaOH and aliquots were removed for protein determination by the bicinchoninic acid (BCA) protein assay. The lipids in each vial were resuspended in 60 μL of hexane and normalized to the total amount of proteins. The lipids were spotted on a Silica Gel 60 F254 thin layer chromatography (TLC) plate. The plate was developed in heptane-diethyl ether-acetic acid (90:30:1) and then exposed to a BAS-MS imaging plate (Fujifilm, Tokyo, Japan) for 5–7 days. The imaging plate was visualized using the FLA-5100 phosphoimager (Fujifilm). The relative intensities of bands corresponding to cholesteryl ester were quantified using Sciencelab ImageGauge 4.0 Software (Fujifilm) or ImageJ (NIH).

**In vivo crosslinking.** ORP1S was subcloned into pEGFP-C1 vector flanked by SalI/KpnI restriction sites. Live cell crosslinking was carried out by exposing GFP or GFP-ORP1S expressing cells with 0.25% paraformaldehyde diluted in PBS for 10 min at room temperature, followed by three rinses with 125 mM glycine in PBS to stop the PFA crosslinking reaction. Cell lysates prepared from treated cells were subject to immunoblot analysis.

**Gel filtration analyses.** The oligomerization states of ORP1-ORD (5 μM) upon ligands binding were analyzed by gel filtration analyses. The Superdex-200 10/300 column was equilibrated with buffer containing 10 mM Tris-HCl, pH 8.0, 150 mM NaCl, and 2 mM DTT and calibrated with molecular mass standards. The ligand (free or incorporated in liposomes) was mixed with ORP1-ORD protein at 4:1 molar ratio, and the mixtures were incubated at 25 °C for 30 min and then subjected to centrifugation. Samples of the ORP1-ORD protein alone, the direct mixtures of protein and ligands, and the supernatants of the protein-liposome mixtures ($400,000 × g$, 1 h) were loaded to the Superdex column and eluted at a flow rate of 0.5 mL min$^{-1}$. All fractions were collected at 0.5 mL each, and aliquots of relevant fractions were all subjected to SDS-PAGE followed by Coomassie Blue staining.

**Crystallography.** Cholesterol, PI(3,4)P₂ or PI(4,5)P₂ was dissolved in chloroform (2 mg/mL in stock), and an aliquot was evaporated using nitrogen gas yielding a thin lipid film. Each ligand (free or incorporated in liposomes) was mixed with ORP1-ORD protein (120 μM) at 4:1 molar ratio, and the mixtures were incubated at 25 °C for 30 min and then subjected to crystallization trials. Crystals were grown using the hanging-drop vapor-diffusion technique by mixing protein-ligand complex (~5 mg/mL) with equal volume of reservoir solution at 18 °C. Crystals of ORP1-ORD (534-950)-cholesterol complex were obtained with reservoir buffer containing 0.1 M Na Cacodylate, pH 6.2, 1.2 M Na Citrate, ORP1-ORD (524-950)-cholesterol complex with reservoir buffer containing 0.1 M NaAc, pH 4.6, 2.0 M Na Formate, and ORP1-ORD (524-950)-PI(4,5)P₂ complex with reservoir buffer containing 0.1 M NaAc, pH 5.7, 1.4 M ammonium tartrate. Crystals of ORP1-cholesterol complexes were equilibrated in a cryoprotectant buffer containing the reservoir buffer supplemented with 25% glycerol and then flash frozen in liquid nitrogen. The crystals of ORP1-ORD-PI(4,5)P₂ complex were cryo-protected in buffer containing 0.1 M NaAc, pH 5.7, 1.8 M ammonium tartrate before snap-frozen.

The diffraction data sets were collected at beamline 17U at Shanghai Synchrotron Radiation Facility and processed with HKL2000[40]. The ORP1-ORD (534-950)-cholesterol structure was solved by molecular replacement using Phaser with yeast Osh3p structure (PDB code: 4IC4) as search model, and structures of ORP1-ORD (524-950)-cholesterol complex and ORP1-ORD (524-950)-PI(4,5)P₂ complex were solved using ORP1-ORD (534-950) as search model[41]. Standard refinement was performed with Coot and PHENIX[42,43]. Ramachandran analyses were carried out using MolProbity[44]. All residues are in the favored and allowed regions of the Ramachandran plot, and none are in disallowed regions. The data processing and refinement statistics are summarized in Table 1. The structural representations were prepared with PyMOL (http://www.pymol.org). The Fo-Fc map was generated using phenix.polder[45].

**Simulation procedures.** The monomeric cholesterol-bound ORP1-ORD was generated from the ORP1-ORD-cholesterol homodimer, where the N-terminal loop and the α1 helix (residues 534–556) adapted from the symmetry-related molecule was linked to the core via a flexible loop (557–565) using MOD-ELLER9v10[46]. The S-(Dimethylarsenic) cysteines (CAS) were substituted by natural cysteines. The program CHARMM36 was then used to add hydrogen atoms, N- and C-terminal patches to the proteins[47], and the CGenFF forcefield was used to add hydrogen atoms to the small molecules[48]. The generated structures were solvated and neutralized in a box with TIP3P model for water at a minimum of 12 Å between the model and the wall of the box[49]. All simulations were performed using the program NAMD 2.9 and CHARMM36 force-fields, and carried out at constant temperature and pressure (NPT) of 300 K and 1 atm[47,50]. The monomeric ORP1-ORD model was subjected to a three-step pre-equilibration of 100 ps, and the last snapshot was chosen as the starting model for 100 ns productive simulation without constraints. The simulation was analyzed with VMD 1.9.3[51]. The behaviors of ORP1-ORD and cholesterol during 100 ns simulation are illustrated in Supplementary Movie 1.

**Liposome association assay**. Some 1 µM wild-type or mutant ORP1-ORD proteins were mixed with 1 mM Rhod-PE-containing liposomes (70% DOPC, 30% PS) in HKM buffer, with or without 0.5% accessible DHE, PI4P or PI(3,4)P$_2$. The mixtures were incubated at 25 °C for 30 min, and then centrifuged at 400,000 × g for 1 h at 20 °C in a rotor (TLA 100.1). The supernatant (200 µL) was recovered, and the liposome pellet, which could be easily visualized owing to the presence of 0.2% Rhod-PE, was washed twice with HKM buffer and resuspended in 200 µL HKM buffer. All samples were analyzed by SDS-PAGE, and the gray value was calculated with ImageJ.

**Kinetic analysis for DHE binding**. All experiments were carried out at 15 °C in a 500 µL reaction mixture containing DOPC/DOPE (2:1) liposomes (200 µM total lipids) doped with 2.5% DNS-PE and 5% DHE in HKM buffer, with or without 4% PI(3,4)P$_2$. The reactions were initiated by addition ORP1-ORD, and the continuous fluorescence changes were recorded and calculated as that in the DHE-binding assay. The initial rates were determined from the linear slope of progress curves obtained, and the experimental data were analyzed with a nonlinear regression model using SigmaPlot 12.0.

The process of binding of a single cholesterol/DHE molecule by ORP1 involves three key steps: (1) association of apo-ORP1 with the DHE-doped liposome, (2) transfer of a DHE molecule from liposome to ORP1, and (3) dissociation of the DHE-bound ORP1 from liposome (Supplementary Fig. 9d).

$$O + L_{(C)} \underset{k_{-1}}{\overset{k_{+1}}{\rightleftharpoons}} O \cdot L_{(C)} \underset{k_{-2}}{\overset{k_{+2}}{\rightleftharpoons}} O_{(C)} \cdot L \underset{k_{-3}}{\overset{k_{+3}}{\rightleftharpoons}} O_{(C)} + L \qquad (1)$$

where O and O$_{(C)}$ correspond to apo-ORP1 and cholesterol-bound ORP1 in solution, L$_{(C)}$ and L correspond to the cholesterol/DHE-doped and cholesterol-extracted (by ORP1) liposomes, O•L$_{(C)}$ is the binary complex formed between O and L$_{(C)}$, and O$_{(C)}$•L is the binary complex of O$_{(C)}$ and L. If the association/dissociation of ORP1 to liposome (steps 1 and 3) are considered to be much faster than the transfer of DHE between liposome and ORP1 (step 2), there is equilibrium as far as O•L$_{(C)}$ and O$_{(C)}$•L are concerned, i.e., when $k_{+3}$ and $k_{-3}$ are sufficient small as not to disturb equilibrium, the equilibrium equations are

$$K_1 = \frac{k_{-1}}{k_{+1}} = \frac{[O][L_{(C)}]}{[O \cdot L_{(C)}]}, \quad K_3 = \frac{k_{-3}}{k_{+3}} = \frac{[O_{(C)}][L]}{[O_{(C)} \cdot L]} \qquad (2)$$

The appearance of DHE bound-ORP1 is given by

$$\frac{d[O_{(C)} \cdot L]}{dt} = k_{+2}\left[O \cdot L_{(C)}\right] - k_{-2}\left[O_{(C)} \cdot L\right]$$
$$= \frac{\left\{k_{+2}\frac{[O]}{K_1} - k_{-2}\frac{[O_{(C)}]}{K_2}\right\}[L_{(C)}]_0}{1 + \frac{[O]}{K_1} + \frac{[O_{(C)}]}{K_2}} \qquad (3)$$

For initial velocity measurements, $[O_{(C)} \cdot L] = [O_{(C)}] = 0$ and $[O]_0 = [O \cdot L_{(C)}] + [O]$ so that

$$v_0 = \frac{k_{+2}\left[L_{(C)}\right]_0[O]}{K_1 + [O]} \qquad (4)$$

The term [O] in Eq. (4) refers to the concentration of free ORP1. Under the present in vitro assay conditions, the ORP1 concentration $[O]_0$ is of the same order of magnitude as the concentration of cholesterol/DHE-doped liposome $[L_{(C)}]_0$. In this case, Eq. (4) can be written as

$$v_0 = k_{+2}[O \cdot L_{(C)}]$$
$$= \frac{k_{+2}}{2}\left\{[O]_0 + [L_{(C)}]_0 + K_1 - \sqrt{\left([O]_0 + [L_{(C)}]_0 + K_1\right)^2 - 4[O]_0[L_{(C)}]_0}\right\} \qquad (5)$$

Similarly, in the presence of PI(4,5)P$_2$/PI(3,4)P$_2$, we have

$$O + L_{(C)}^* \underset{k_{-1}^*}{\overset{k_{+1}^*}{\rightleftharpoons}} O \cdot L_{(C)}^* \underset{k_{-2}^*}{\overset{k_{+2}^*}{\rightleftharpoons}} O_{(C)} \cdot L^* \underset{k_{-3}^*}{\overset{k_{+3}^*}{\rightleftharpoons}} O_{(C)} + L^* \qquad (6)$$

$L_{(C)}^*$ corresponds to the liposome doped with cholesterol/DHE and PI(4,5)P$_2$/PI(3,4)P$_2$, L* is the liposome when cholesterol extracted by ORP1 (Supplementary Fig. 9e). The initial velocity is given by

$$v_0^* = k_{+2}^*[O \cdot L_{(C)}^*] =$$
$$\frac{k_{+2}^*}{2}\left\{[O]_0 + [L_{(C)}^*]_0 + K_1^* - \sqrt{\left([O]_0 + [L_{(C)}^*]_0 + K_1^*\right)^2 - 4[O]_0[L_{(C)}^*]_0}\right\} \qquad (7)$$

where

$$K_1^* = \frac{k_{-1}^*}{k_{+1}^*} = \frac{[O][L_{(C)}^*]}{[O \cdot L_{(C)}^*]} \qquad (8)$$

**PIP delivery assay**. ORP1 (final concentration of 10 µM) was incubated at 25 °C with the donor liposomes (DOPC/DOPE at 2:1 molar ratio; final concentration of 1 mM) doped with 10% PIP. After 30 min, the mixture was centrifuged at 400,000 × g for 1 h. Then, the supernatant (containing ORP1, final concentration of 3 µM) was mixed with acceptor liposomes (DOPC/DOPE at 2:1 molar ratio; final concentration of 250 µM) in HKM buffer. NBD-PH$_{FAPP}$ (final concentration of 250 nM) was also added, and the PIP delivery was monitored at 525 nm (bandwidth 10 nm) upon excitation at 490 nm (bandwidth 10 nm).

**Statistical analyses**. The data are presented as the mean ± s.e.m. or s.d.; Statistical analyses were performed with one-way ANOVA.

**Reporting summary**. Further information on experimental design is available in the Nature Research Reporting Summary linked to this article.

## Data availability
Data supporting the findings of this manuscript are available from the corresponding authors upon reasonable request. A reporting summary for this Article is available as a Supplementary Information file. The coordinates and structure factors have been deposited in the Protein Data Bank with accession numbers 5ZM5 and 5ZM7 for ORP1-ORD in complex with cholesterol and 5ZM6 for ORP1-ORD in complex with PI(4,5)P$_2$. The source data underlying Figs. 1b-d, 2b, 2d-k, 3b-d, 5b, 6a-b, 7a, 7c-d, 8a-b, 8f-g, 9b and Supplementary Figs. 2a-b, 3b, 4a-b, 4e, 6a, 7, 8b, 9b-c, 10b, 11a-b, 12a-b are provided as a Source Data file.

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

## Acknowledgements

We thank the Shanghai Synchrotron Radiation Facility and the Tsinghua University Branch of China National Center for Protein Sciences (Beijing) for providing the facility support; J.-W.W. is supported by grants from National Key R&D Program of China (2016YFA0502004) and National Natural Science Foundation of China (91754201 and 31621063). H.Y. is supported by the National Health and Medical Research Council (NHMRC) of Australia grants (1041301 and 1141939). H.Y. is an NHMRC Senior Research Fellow (1058237).

## Author contributions

The X-ray crystallography, bioinformatics and structural modeling were carried out by J. D., H.W., J.W., C.L., X.C., Z.Z., and Z.L. X.D. performed the cell biology experiments with advice from A.J.B. H.W. and J.D. conducted lipid binding and transport experiments with advice from J.-W.W. and L.Y. J.-W.W. and H.Y. conceived the project and wrote the manuscript together with all other authors.

## Additional information

**Competing interests:** The authors declare no competing interests.

