## [Peer Review File · Nature Communications]

Reviewers' Comments:

Reviewer #1:

Remarks to the Author:

In this manuscript, Wu and co-workers use a thoughtful combination of approaches (in vitro reconstitution, cellular microscopy, x-ray crystallography, molecular dynamics) to investigate how lipid transport by ORP1 is regulated.

The main finding of this work is that two PIP2 lipids, PI(4,5)P2 and PI(3,4)P2, allosterically enhance cholesterol transport when present in either the donor or acceptor membrane. This finding is supported by cellular experiments showing the involvement of PI3KC2B in ORP1-mediated cholesterol transport, and a possible structural mechanism is proposed from the analysis of X-ray structures and MD simulations on cholesterol-bound and PIP2 bound ORP1 structures. To my understanding, this mechanism of allosteric regulation of lipid transport at membrane contact sites by phosphoinositides can be considered a new finding, and it could warrant publication on Nature Communications.

On the other hand, before publication, the authors must address two main apparent inconsistencies that raise doubts on the quality of the experiments and on the validity of the final model proposed for lipid transport by ORP1. This stems in particular from the unexpected behavior of ORP1 with respect to PI(4)P binding and transport, and I find somewhat appalling that the authors do not discuss this issue at length in the Discussion section. In detail:

1. In a recent article by the Ridgeway lab (Zhao and Ridgeway, Cell Reports, 2017), ORP1 is shown to bind and extract PI(4)P. This is at odds with the finding presented in this work (Fig.1d and Extended Data Fig.2d) where the authors show that PI(4)P is not transported at all by ORP1

2. In Fig.1b, in what the authors call "lipid binding/extraction" assay, PI(4)P is one of the more markedly bound/extracted PIPs. Yet, according to the subsequent transport assay, PI(4)P is not transported to the donor liposomes.

A possible explanation of these findings that is consistent with the data presented here and in the Ridgeway paper is that PI(4)P is (1) extracted by ORP1 but not transported or (2) extracted by ORP1 and then transported back only to the donor membrane. However, from a cellular point of view, this seems counterintuitive, since both mechanisms would strongly hinder the capability of ORP1 to transport cholesterol.

In any case, the authors should understand and convincingly describe the lipid binding, extraction and transport of PI(4)P by ORP1 in more detail before this work can be considered for publication. In simpler terms, if ORP1 is a cholesterol transporter that is allosterically enhanced by PI(4,5)P2 and PI(3,4)P2, why is it binding and extracting almost all the accessible PI(4)P (Fig.1b)?

To this end, there are several experiments that the authors should try to do in order to tackle this incongruence:

1. The authors only work with a truncated version of ORP1S for their transport assay (the effect of truncation is only shown for the binding/extraction assay). They should investigate whether the full-length version has a different transport profile.

2. The authors should try changing the size and the lipid composition of the various liposomes used as donors and acceptors. One possibility is that PI(4)P is extracted but not released because the acceptor membrane is not able to accept PI(4)P. To overcome this, the author could use acceptor liposomes of smaller radius and/or higher lipid unsaturation.

This should also help alleviate another important problem observed by the authors in that, in their transport assay, cholesterol transport is the same with or without ORP1.

3. If the transport of cholesterol is direct between LELs and the ER (as expected for ORP1L because of the VAP-binding FFAT domain), the authors should also test major ER lipids such as PI and PA.

Other minor points also need to be addressed:

1. The pdb files of the X-ray structures are not available and it is thus difficult to evaluate some of the claims made in the second part of the manuscript by simply looking at the figures. To this end, making available the electron density maps for the review step would be beneficial. In any case, the interpretation of the oligomeric states found in the X-ray is dubious at best, since it is completely unclear (and not investigated) whether these oligomeric states and interactions represent physiological oligomeric states or they are simply a consequence of crystallization conditions.
2. When discussing the in vitro reconstitution assays, I would strongly suggest to describe those assays in more detail directly in the main text, rather than only in the Methods section: these assays are not standard for a wide audience.
3. For the in vitro assays (both binding/extraction and transport), the time traces of the fluorescent signal upon protein addition are generally more informative than the normalized plots provided in the main text, and should be used directly in the main text.

Reviewer #2:

Remarks to the Author:

In this study, Dong, et al. utilized an array of biochemical and structural approaches to determine the function of the ORP1-ORD, the lipid binding domain in ORP1L and ORP1S. They propose that the binding and transfer of sterols by the ORP1-ORD is allosterically activated by PI-3,4-P2 and PI-4,5-P2 at the limiting membrane of the LEL. However, they suggest that the OHD does not bind PIPs. The study is significant for being the first to solve the structure of a mammalian ORD in association with cholesterol and a PIP. However, the study is very complex, leading to superficial description of experiments, results and conclusions, and relies substantially on data from lipid extraction and transfer assays that are not designed to show this activity. This and other major concerns are outlined below.

1. The authors acknowledge that the assays shown in Fig. 1b cannot resolve binding/extraction. In reality, all that these assays can measure is liposome binding and not the ability of the ORP1-ORD to extract either DHE or PIPs. They repeatedly refer to this in the text as extraction, which is not the case.
2. The interpretation of Fig. 1C is problematic. This is referred to as a DHE transport assay based on the premise that acceptor liposomes with different PIPs stimulate the removal of DHE (loss of FRET signal) from the donor. This assay does not measure DHE transport since the arrival of DHE at the acceptor is not measured; it is only assumed that DHE arrives at the PIP containing liposome. Why wasn't DNS-PE included in the acceptor liposome instead to quantitate the arrival of DHE into that liposome (ie. an increase in the FRET signal)? This was the design of the transport assays described in the Mesmin Cell paper in 2013. The assays described here are not appropriate to measure transfer. Unfortunately these assays are used throughout the study.
3. In Fig. 1b, it is not clear how the NBD-PH signal is quenched once it dissociates from the liposome. Is this an intrinsic property of the NBD-PH or is a quencher or FRET partner missing?
4. In Fig. 1c it is indicated that acceptor liposomes without DHE were tested but this is not shown in any figures. It is possible that the lack of PIP transfer in the presence of DHE could be due to more efficient binding of the sterol compared to PIPs.
5. In Fig. 1, Osh4p is used as a control. It is clearly much more active than ORP1-ORD but this is not commented on. Is ORP-ORD just a poor sterol binder?

6. In Fig. 2i and 3j, the assay does not measure transport but only binding or extraction of DHE from the donor liposome; the assay is not designed to register DHE arrival at the acceptor liposome (see #2 above).

7. In Fig. 4 and extended data 6, it is clearly shown that the ORP1-ORD binds very tightly to PI-3,4-P2 containing liposomes using a sedimentation assay. This being the case, it is highly unlikely that the ORP1-ORD will dissociate significantly from liposomes containing this lipid whether it is bound to sterol or not. There is no evidence provided that PI-3,4-P2 loosens its grip on the ORP1-ORD when sterol is bound. This puts into question the model proposed in which PI-3,4-P2 serves as an allosteric activator.

8. The authors should show the change of level of PI-3,4-P2 level in the PI3KC2beta knockdown cells, either by mass spec, HPLC or antibody-based methods. Plus, they might try over-expression of PI3KC2beta in HeLa cells and assess its impact on cholesterol transport from LELs.

9. Related to the model in Fig. 5, what is 'basal extraction' and is it any more efficient than stimulated extraction. If PI-3,4P2 is a strong binder of the ORP1-ORD, perhaps it is an inhibitor of basal sterol extraction (although Fig. 2i, might suggest otherwise).

10. There is some concern about the nature of the ORP1-ORD used in the study. These truncated versions start at 524 and 534, while ORP1S starts at amino acid 514. While this might be an insignificant structural truncation, there is the possibility that removal of these 10-20 amino acids affects the behavior of the ORP1-ORD, perhaps causing the unusual higher order structures described in Fig. 3. The authors also do not address whether ORP1S multimerizes (one would predict this based on their structural data), and it should be easy to test in cells. The authors could have at least commented on this point.

11. The study is focused on the role of ORP1-ORD in sterol removal from the LEL. This form is very similar to the native ORP1S. However, most studies have reported that ORP1L is strongly localized to LELs and mediates cholesterol export (which they seem to confirm in Fig. 2). The overall tone of the manuscript is that the ORD (ie. ORP1S) is also involved in cholesterol transport from the LEL (the model in Fig5 shows the ORD/ORP1S), but this is not directly addressed in any of the experiments described in Fig. 2 or elsewhere. ORP1S is not known to associate with the LEL, instead it appears to be in the cytoplasm. If their model of LEL targeting by interaction with PI-3,4-P2 is correct, one would assume that ORP1S would be present on the LEL. This could have been addressed by re-expressing ORP1S in the DKO HeLa cells to see if it was sufficient to restore cholesterol trafficking to the ER.

12. Lysosomal cholesterol has the potential to activate mTORC1. In ORP1 KO cells, there is more mTORC1 activity as shown in Fig. 2j and k. The authors should check whether deficiency of ORP1 affects autophagy?

Minor points:

Figure 3I—"N-terminla" spelled incorrectly

Figure 4J—not clear which curve refers to the presence or absence of PI45P2

Line 69, should say 'we expressed' not 'subcloned'.

Reviewer #3:

Remarks to the Author:

The non-vesicular lipid transport coupled by phosphoinositide-counter transport are currently a major focus of interest in lipid biology. Many ORPs were suggested to transport sterol or PS

against their concentration gradients using a PI4P gradient as an energy source. This work reveals a distinct mechanism from previous findings in that ORP1 mediates a down-hill transport of cholesterol under allosteric regulation of PIPs, which explains the specific role of ORP1 in sterol transport at the LEL and ER interface. The authors unraveled the ill-defined features of ORP1 by various biochemical and cell-biology approaches, which are convincing in overall.

I favor publication of the manuscript in condition that the following points are addressed in the revised version.

ORP1 extracts PI4P and other PIPs well from the liposomes (Fig 1b). However, authors observed that ORP1 does not transport any PIPs at all. This finding is quite intriguing considering the conservation of PI4P transport in other ORPs. This feature needs to be confirmed using the acceptor liposomes containing PS to examine the dependency of membrane association for PIP transport. In addition, DHE transport (Fig 1c) should be tested using the liposomes containing PS in the acceptor liposomes. Another possibility is that for the PI4P transport ORP1 might require the presence of PI(3,5)P2 or PI(4,5)P2 in the acceptor membrane, which might help PI4P release by opening the N-terminal lid. Did you examine the PIP2-dependent transport of PI4P?

One of the most important data is the crystal structure of PI(4,5)P2 bound ORP1. The orientation of PI(4,5)P2 head group is different compared to the PI4P of other ORPs, which was explained by the rotatable bonds in the inositol ring. However, the electron densities of the head group and other region of the ligand are not clearly resolved. The authors should provide unbiased omit maps (Fo-Fc) of the PI(4,5)P2 in several different orientations for the quality check of the ligand structures.

The binding mode of PI(4)P are well preserved in all reported ORP structures and the strict conservation of the key residues of the head-group binding site suggests PI(4)P binding is a common feature of ORPs. Authors should provide a figure showing the comparison of the PIP binding of ORPs in detail (eg. Structure superposition of PI(4,5)P2 and PI(4)P including key residues). What is the unique structural features that allow the binding of PI(4,5)P2 in ORP1 compared to other ORPs?

The labels in the figures are too small to read.

Reviewer #4:

Remarks to the Author:

This interesting study focuses on the structure and the function of the lipid transfer protein ORP1L, and in particular its C-terminal part, which corresponds to the ORD domain, (i.e. without the domains known to bind specifically to membranes). The authors suggest that ORP1L is a cholesterol transporter and attempt to demonstrate that it participates in the release of cholesterol from late endosomes/lysosomes (LELs) and to its transfer to the ER. A recent study (Zhao et al. Cell Rep 2017) has already described ORP1L as a lipid transfer protein involved in the exit of chol from LEL, but as a potential chol/PI4P exchanger. However, that study did not provide a consistent mechanism for the PI4P metabolism part. Here the authors suggest that ORP1 does not transfer PIPs, but that PIP2 acts as an allosteric activator to facilitate cholesterol transfer. The authors use in vitro approaches (lipid transfer between liposomes by purified proteins) combined with structural approaches and molecular dynamics simulation. They provide the crystal structure of the ORD in complex with cholesterol or with PI(4,5)P2. In addition, the authors also measure the effect of ORP1L/S KO on the distribution and metabolism of intracellular cholesterol, and on the activation of mTORC1, which depends on cholesterol and PI(3,4)P2 levels in LELs, as previously demonstrated. Although the role of this protein seems modest or even dispensable for cell function and LEL function, the study has the merit of providing new, interesting and unexpected results for a protein from this family. Importantly however, some key experiments are not being carried out

as they should be, and some controls and information are missing. They are necessary for a good understanding of the manuscript and for the relevance of the study. So I require a major revision for this manuscript. Below are my detailed comments.

1) PIP binding/extraction and PIP transfer. The authors make the serious mistake of using the FAPP PH domain as a probe to measure the ORD's binding to PIPs and its PIP transfer activity. It makes no sense to use a probe that only binds specifically to PI(4)P to measure the transport of other PIPs, or the ORD binding to other PIPs, since these other PIPs are not recognized by the FAPP PH domain. The experiments in Fig 1b, 1d, Extended fig 2b, 2d, 5e must be repeated with specific probes for each PIP studied. A biochemical validation of these probes will probably be necessary, if this has not been done previously, before using them for binding or transfer experiments by an ORD (Moser von Filseck et al. 2015 Nat Com has characterized biochemically the FAPP-PH). This will apparently require a lot of work, but it is crucial for this study to demonstrate that the ORD is not able to extract PIPs.

2) Activation of mTORC1. A 2-fold increase in S6K phosphorylation is not a "dramatic" increase as written on line 128. It has been shown by Castellano BJ et al. 2017 Science, that in conditions of NPC1 null or inhibited, there is a constitutive activation of mTORC1. Here, the ORP1 KO in the condition without FBS has no visible effect on mTORC1 activation. In addition, to prove that LELs are involved, the authors must manipulate the LDL. Thus, it would be interesting to compare for example the effect of the FBS with that of the lipoprotein-deprived serum. Concerning PI3KC2beta, the paper Marat AL et al. 2017 Science clearly shows a phosphorylation of S6K in conditions without FBS upon PI3KC2beta silencing. Therefore, the authors should tone down their statements on the role of ORP1L in the activation of mTORC1.

3) There is no evidence in this paper that PI3KC2beta siRNA reduces the level of PI(3,4)P2 in LELs.

4) I see the arguments suggesting that ORP1L transfers cholesterol from LELs to ER, and not from ER to LELs, but the Extended Data Fig 3f suggest that the ORD sterol transfer rate is very, very slow. Such a weak activity cannot reasonably play an important role in the release of cholesterol from LEL, which is the main entry point for cholesterol into the cell. Moreover, the transfer is a little more effective when the PIPs are in the accepting liposomes (fig 1c). In fact, the results obtained with the ORD domain in Figure 1 do not prove that ORP1L (i.e. the entire form) does the same. For lipid binding and transfer, it has been shown that the interaction of some ORPs, including ORP1L, with VAP influences lipid binding (Weber-Boyvat M et al. CMLS 2015; Mesmin B et al. Cell 2013). The authors should perform similar lipid transfer experiment with the full-length protein ORP1L, and by adding VAP. The membrane binding of the PH domain and the binding to VAP could accelerate the transfer.

5) Molecular dynamics. It is not clear whether the sterol molecule is in continuous rotation in the ORD tunnel or whether it is transient due to the initial state with the lid open. The question is: does the rotation stop when the lid closes and is it possible to quantify the fluctuations (rotations) of cholesterol in the tunnel over time (ie. during the first 20-30 ns, to do the correlation with the movements of the lid (measured in the Extended Data Fig 5a). This would strengthen the data suggesting that the molecules in the crystal structure are in an intermediate state.

6) For the gel filtration, the sentence lines 182-183 is not clear. Moreover, the protocol is also unclear. What are the concentrations of cholesterol and protein used, the incubation conditions? It is also wrong to write that there is no change in the gel filtration profile of fig 3d upon chol addition: The comparison between the black and dark blue lines shows a strong shift towards the monomer. Can the authors provide an explanation?

7) For oligomers, the authors do not discuss their possible physiological relevance, or whether they even exist in the cell. It is not clear if the dimer and trimer (fig 3 e, f) keep the right orientation of

the basic patches (all on the same side)? In the tetramer when the ORD is bound to PI(4,5)P2, the lipid appears to be trapped between two proteins, which seems unlikely in reality. Can the authors comment?

8) Is the fact that the sn-1 chain of PIP2 enters into the ORD tunnel compatible with the main message of the manuscript, ie. that PIP2 stimulates cholesterol transport by the ORD? I could understand if it stimulates the release of cholesterol from the protein by a competition mechanism, but the role of allosteric activator is not very intuitive considering that both ligands occupy the same binding site (at least part of it). An interesting experiment would be to measure the binding of ORD to liposomes containing both DHE and PI(3,4)P2. In this case, is there a reduction in membrane binding compared to when only PI(3,4)P2 is present (fig 4e)? This would help to understand the mechanism. Moreover, the effect of PI4P in this figure is disturbing. The authors discuss little about this effect in the manuscript.

9) Although the effect on Filipin is clear in Figure 5, how do the authors explain that mutants that are supposed to do less interaction with membranes are much more enriched in LELs? This is particularly striking with mutants of basic patches: they are completely recruited to late endosomes as compared to the WT.

Minor points:

10) Related to Fig 2, the authors should tone down their statements: ORP1L is not "required" for the removal of cholesterol from LELs as it is written line 114. ORP1L has a modest contribution at best (compare the effect of U18666A or NPC1 KO to the effect of the KO of ORP1L and S).

11) In the end of the introduction the authors mention the study by Zhao et al. Cell Rep 2017, but do not mention another study that suggests that in the absence of LDL-cholesterol, ORP1L binds VAP, and this interaction promotes cholesterol transfer from the ER to MVBs to support ILV formation (Eden ER et al. 2016 Dev Cell). So that would work the other way round from the model presented by the authors in the present manuscript. It would be interesting to confront that study with the current one.

12) Fig 1b shows the % of "accessible" DHE. It should also be written in line 75.

13) In Fig 1, it is sometimes written 460 or 490 for NBD excitation.

14) Line 107: replace « Extended Data Fig 3c » for « Extended Data Fig 3d ».

Point-by-point responses to reviewers' comments:

#1 (Remarks to the Author):

In this manuscript, Wu and co-workers use a thoughtful combination of approaches (in vitro reconstitution, cellular microscopy, x-ray crystallography, molecular dynamics) to investigate how lipid transport by ORP1 is regulated.

The main finding of this work is that two PIP2 lipids, PI(4,5)P2 and PI(3,4)P2, allosterically enhance cholesterol transport when present in either the donor or acceptor membrane. This finding is supported by cellular experiments showing the involvement of PI3KC2B in ORP1-mediated cholesterol transport, and a possible structural mechanism is proposed from the analysis of X-ray structures and MD simulations on cholesterol-bound and PIP2 bound ORP1 structures.

To my understanding, this mechanism of allosteric regulation of lipid transport at membrane contact sites by phosphoinositides can be considered a new finding, and it could warrant publication on Nature Communications.

We greatly appreciate your positive comments on our work.

On the other hand, before publication, the authors must address two main apparent inconsistencies that raise doubts on the quality of the experiments and on the validity of the final model proposed for lipid transport by ORP1. This stems in particular from the unexpected behavior of ORP1 with respect to PI(4)P binding and transport, and I find somewhat appalling that the authors do not discuss this issue at length in the Discussion section. In detail:

1. In a recent article by the Ridgeway lab (Zhao and Ridgeway, Cell Reports, 2017), ORP1 is shown to bind and extract PI(4)P. This is at odds with the finding presented in this work (Fig.1d and Extended Data Fig.2d) where the authors show that PI(4)P is not transported at all by ORP1

2. In Fig.1b, in what the authors call "lipid binding/extraction" assay, PI(4)P is one of the more markedly bound/extracted PIPs. Yet, according to the subsequent transport assay, PI(4)P is not transported to the donor liposomes.

A possible explanation of these findings that is consistent with the data presented here and in the Ridgeway paper is that PI(4)P is (1) extracted by ORP1 but not transported or (2) extracted by ORP1 and then transported back only to the donor membrane. However, from a cellular point of view, this seems counterintuitive, since both mechanisms would strongly hinder the capability of ORP1 to transport cholesterol.

In any case, the authors should understand and convincingly describe the lipid binding, extraction and transport of PI(4)P by ORP1 in more detail before this work can be considered for publication. In simpler terms, if ORP1 is a cholesterol transporter that is allosterically enhanced by PI(4,5)P₂ and PI(3,4)P₂, why is it binding and extracting almost all the accessible PI(4)P (Fig.1b)?

Thank you very much for your advice. Zhao and Ridgeway (*Cell Reports*, 2017) have shown that ORP1 can bind and extract PI4P, but did not assess whether ORP1 can transport PI4P between membranes. Consistently, we also found that ORP1-ORD could bind and extract PI4P (Figs. 1b and 8a). However, our *in vitro* biochemical analyses showed that ORP1 cannot transport PI4P (Fig. 1d). These results indicate that ORP1 may be incapable of unloading PI4P to membrane. To verify this hypothesis, we performed delivery assays for PI4P and PI(4,5)P₂ (see METHODS on pages 39-40 in the revised manuscript). Briefly, the lipid-bound ORP/Osh protein was mixed with liposome (the acceptor). If the lipid can be released from protein and further embedded into liposome, the fluorescent signal of NBD-PH_{FAPP} would change. As shown in new Supplementary Fig. 12a, ORP1-ORD cannot deliver PI4P or PI(4,5)P₂ to liposomes, regardless of the presence of DHE. Consistent with the cholesterol-PI4P counter transport mechanism, Osh4p can weakly deliver PI4P and PI(4,5)P₂, but only the PI4P delivery can be enhanced by DHE/cholesterol embedded in the acceptor liposome. Therefore, though ORP1 can bind and extract PI4P, it cannot transport PI4P because the ORP1-bound PI4P cannot be released from ORP1-ORD and unloaded to liposome/membrane.

As you mentioned, since ORP1-ORD can bind and extract PI4P yet cannot release and deliver PI4P, the DHE transport by ORP1-ORD would be hindered by PI4P. In the DHE transport assays, the concentration of ORP1-ORD we used is 0.1 μM, which allows us to show the PI(4,5)P₂/PI(3,4)P₂-stimulation on the initial velocity of DHE transfer (Fig. 1c). By contrast, PI4P displayed weak inhibition. To clearly examine the effect of PI4P, we increased the concentration of ORP1-ORD to 0.5 μM. As shown in new Supplementary Fig. 12b, PI4P did inhibit DHE transfer by ORP1.

As to whether PI4P hinders cholesterol transport *in vivo*, their subcellular distribution and abundance must be taken into consideration. ORP1 is localized to LELs and mediates cholesterol export. LDL is sorted and delivered to LELs through the receptor-mediated endocytosis, and the released free cholesterol then reaches the limiting membrane of LELs. However, PI4P on the LEL limiting membrane is of low abundance (Choy CH, *et al. Bioessays*. 2017; Balla T. *Physiol Rev*. 2013). It is hard to compare exactly the ratio of cholesterol and PI4P, but we estimate that the level of cholesterol is 200-500 times more than that of PI4P (Balla T. *Physiol Rev*. 2013; Hullin-Matsuda F, *et al. Semin Cell Dev Biol*. 2014; van Meer G, *et al. Nat Rev Mol Cell Biol*. 2008). Thus, PI4P might not hinder cholesterol binding and transport by

ORP1 *in vivo*. As suggested, we have added discussion about PI4P on page 23 in the revised manuscript.

To this end, there are several experiments that the authors should try to do in order to tackle this incongruence:

1. The authors only work with a truncated version of ORP1S for their transport assay (the effect of truncation is only shown for the binding/extraction assay). They should investigate whether the full-length version has a different transport profile.

According to your suggestion, we investigated the transport activities of ORP1S (514-950) and the longer truncation (524-950). As shown in new Supplementary Fig. 3a and d, these two ORP1-ORD fragments displayed similar transport profiles for DHE and PIPs as the short truncation (534-950) which we used in most biochemical analyses. Thus, the N-terminal truncation did not affect the lipid binding and transport of ORP1-ORD. We also examined the lipid binding and transfer profiles of full-length ORP1L (please see new Fig. 3 and results on pages 8-9 in the revised manuscript).

2. The authors should try changing the size and the lipid composition of the various liposomes used as donors and acceptors. One possibility is that PI(4)P is extracted but not released because the acceptor membrane is not able to accept PI(4)P. To overcome this, the author could use acceptor liposomes of smaller radius and/or higher lipid unsaturation.

This should also help alleviate another important problem observed by the authors in that, in their transport assay, cholesterol transport is the same with or without ORP1.

Thank you for your advice. To examine the effect of liposome size, we performed the PI4P transport assay using acceptor liposomes of smaller radius (50, 75 and 100 nm). As shown below, the size of acceptor liposome did not affect the PI4P transfer by ORP1. Importantly, Osh4p can transfer PI4P in our *in vitro* assays, with or without DHE embedded in the acceptor liposomes (black lines in Fig. 1d and new Supplementary Fig. 3c.) Therefore, we believe that lipid composition might not interfere with the PI4P transport by ORP1. We put the data here for your information only.

PI4P transport assays using acceptor liposomes (L_A liposomes) of different radius.

Please note that the initial velocities of DHE transport by ORP1 shown at the bottom of Fig. 1c were subtracted by that without ORP1. As we mentioned above, the concentration of ORP1-ORD used in the DHE transport assays is $0.1 \mu\text{M}$, which allows us to show the $\text{PI}(4,5)\text{P}_2/\text{PI}(3,4)\text{P}_2$ -stimulation on the initial velocity of DHE transfer. When the concentration of ORP1-ORD is increased to $0.5 \mu\text{M}$, the difference of DHE transport with or without ORP1 is obvious (new Supplementary Fig. 12b, grey trace/bar vs marine blue trace/bar).

3. If the transport of cholesterol is direct between LELs and the ER (as expected for ORP1L because of the VAP-binding FFAT domain), the authors should also test major ER lipids such as PI and PA.

This is a good point. We carried out DHE/cholesterol transport assay using the acceptor liposomes supplemented with 4% PI or PA, the same amount as PIP_2 used in Fig. 1c. As shown below, PI or PA had little effect on cholesterol transport by ORP1, consistent with the reported lipid-protein overlay results that ORP1-ORD did not interact with PI or PA (Fairn GD and McMaster CR. *Biochem J.* 2005). This is for your information only.

DHE transport assays with PI, PA or $\text{PI}(3,4)\text{P}_2$ incorporated in the acceptor liposomes.

Other minor points also need to be addressed:

1. The pdb files of the X-ray structures are not available and it is thus difficult to evaluate some of the claims made in the second part of the manuscript by simply looking at the figures. To this end, making available the electron density maps for the review step would be beneficial. In any case, the interpretation of the oligomeric states found in the X-ray is dubious at best, since it is completely unclear (and not investigated) whether these oligomeric states and interactions represent physiological oligomeric states or they are simply a consequence of crystallization conditions.

As suggested, we uploaded the electron density maps and PDB files of three ORP1 structures for your reference.

The oligomeric states of ORP1-ORD in isolation and in complex with cholesterol or PI(4,5)P₂ have been analyzed by gel filtration chromatography. The ligand-free and cholesterol-bound ORP1-ORD proteins exist in three different states (monomer, dimer and trimer), regardless of the complex preparation procedures (Fig. 5a and Supplementary Fig. 6a). On the other hand, ORP1-ORD will be recruited to the PI(4,5)P₂-incorporated liposomes, but can tetramerize upon free PI(4,5)P₂ binding (Supplementary Fig. 10b). To further test whether ORP1-ORD forms oligomers *in vivo*, we transiently transfected HEK293 cells with GFP empty vector or GFP-tagged ORP1S (514-950), and then briefly exposed cells to the crosslinker paraformaldehyde (PFA, 0.25% v/v) prior to lysis for immunoblotting analysis using a monoclonal GFP antibody. In the presence of PFA, GFP-ORP1S (~70 kDa) appeared to form oligomers with the sizes ranging from 150 kDa to 220 kDa (new Fig. 5b). These results suggest that the oligomeric states (at least the monomer, dimer and trimer) represent physiological states of ORP1. We've added the results and methods on pages 11 and 34 in the revised manuscript.

2. When discussing the in vitro reconstitution assays, I would strongly suggest to describe those assays in more detail directly in the main text, rather than only in the Methods section: these assays are not standard for a wide audience.

Thank you. We've described these binding and transport assays in more detail in the main text (pages 5-6 in the revised manuscript).

3. For the in vitro assays (both binding/extraction and transport), the time traces of the fluorescent signal upon protein addition are generally more informative than the normalized plots provided in the main text, and should be used directly in the main text.

As suggested, we moved the time traces of the fluorescent signal to the main text (new Figs. 1 and 3).

Reviewer #2 (Remarks to the Author):

In this study, Dong, et al. utilized an array of biochemical and structural approaches to determine the function of the ORP1-ORD, the lipid binding domain in ORP1L and ORP1S. They propose that the binding and transfer of sterols by the ORP1-ORD is allosterically activated by PI-3,4-P2 and PI-4,5-P2 at the limiting membrane of the LEL. However, they suggest that the OHD does not bind PIPs. The study is significant for being the first to solve the structure of a mammalian ORD in association with cholesterol and a PIP. However, the study is very complex, leading to superficial description of experiments, results and conclusions, and relies substantially on data from lipid extraction and transfer assays that are not designed to show this activity. This and other major concerns are outlined below.

We greatly appreciate your constructive and insightful comments.

1. The authors acknowledge that the assays shown in Fig. 1b cannot resolve binding/extraction. In reality, all that these assays can measure is liposome binding and not the ability of the ORP1-ORD to extract either DHE or PIPs. They repeatedly refer to this in the text as extraction, which is not the case.

Thank you for your advice. The assays shown in Fig. 1b and Supplementary Fig. 2a were referred to as “extraction assays” in several papers (de Saint-Jean M, *et al. J Cell Biol.* 2011; Moser von Filseck J, *et al. Nat Commun.* 2015; Moser von Filseck J, *et al. Science* 2015). We did notice that the fluorescent signal changes in these assays cannot tell whether the lipid is extracted from liposome or not, and thereby referred to them as the “binding/extraction assays”. We had performed the liposome/membrane association assay to further resolve binding/extraction, and found that DHE and PI4P can be extracted by ORP1-ORD while PI(3,4)P₂/PI(4,5)P₂ can recruit ORP1 to liposome/membrane (Fig. 8a). As suggested, we now referred to these assays as the “binding assays” in the revised manuscript.

2. The interpretation of Fig. 1C is problematic. This is referred to as a DHE transport assay based on the premise that acceptor liposomes with different PIPs stimulate the removal of DHE (loss of FRET signal) from the donor. This assay does not measure DHE transport since the arrival of DHE at the acceptor is not measured; it is only assumed that DHE arrives at the PIP containing liposome. Why wasn't DNS-PE included in the acceptor liposome instead to quantitate the arrival of DHE into that liposome (ie. an increase in the FRET signal)? This was the design of the transport assays described in the Mesmin Cell paper in 2013. The assays described here are not

appropriate to measure transfer. Unfortunately these assays are used throughout the study.

This is a good point. Please note this DHE transport assay with DNS-PE embedded in the donor liposome was used in the NC paper by Guillaume Drin's lab (Moser von Filseck J, *et al. Nat Commun.* 2015). Both Drs. Guillaume Drin and Moser von Filseck J are co-authors of the Cell paper by Bruno Antonny's lab (Mesmin B, *et al. Cell.* 2013). Importantly, the concentration of protein we used in this DHE transport assay is 0.1 μM , which is much lower than the DHE concentration (10 μM). The amount of DHE transported by ORP1 is more than 0.1 μM , especially in the presence of PI(3,4)P₂/PI(4,5)P₂ (Fig. 1c). Thus, we believe that DHE is indeed transported to the acceptor liposome.

As suggested, we carried out DHE transport assay with DNS-PE incorporated into the acceptor liposome. As shown below, when the acceptor liposomes were free of any PIPs, ORP1-ORD transports DHE/cholesterol slowly. The DHE transport was markedly enhanced when the acceptor liposomes were supplemented with PI(3,4)P₂ or PI(4,5)P₂. These results are consistent with that from the DHE transport assays with DNS-PE in the donor liposome. We put the data below for your information only since we run out of space in Fig. 1.

DHE transport assays with DNS-PE incorporated in the acceptor liposome. The donor liposome (L_A, DOPC/DOPE, 2:1; total lipids, 200 μM) was doped with 5% DHE, and the acceptor liposome (L_B, DOPC/DOPE, 2:1; total lipids, 200 μM) contained 2.5% DNS-PE and 4% indicated PIP. The DHE transport was followed by measuring the dansyl signal at 525 nm (bandwidth 10 nm) upon DHE excitation at 310 nm (bandwidth 10 nm).

3. In Fig. 1b, it is not clear how the NBD-PH signal is quenched once it dissociates from the liposome. Is this an intrinsic property of the NBD-PH or is a quencher or FRET partner missing?

NBD is a sensitive fluorescent probe and its photophysical characteristics depend strongly on the environment. It has a good quantum yield in lipid bilayers while being weakly fluorescent in water. So once NBD-PH_{FAPP} is competed off by ORP/Osh proteins and dissociates from the liposome, the NBD signal would be quenched. The NBD-labeled derivatives have been widely used as fluorescent membrane probes (Amaro M, *et al. Phys Chem Chem Phys*. 2016). We've added this on page 5 in the revised manuscript.

4. In Fig. 1c it is indicated that acceptor liposomes without DHE were tested but this is not shown in any figures. It is possible that the lack of PIP transfer in the presence of DHE could be due to more efficient binding of the sterol compared to PIPs.

We accepted your advice and have added the data of PIP transport without DHE in the acceptor liposome (new Supplementary Fig. 3c). The results showed that ORP1-ORD did not transport any PIP when DHE is absent.

5. In Fig. 1, Osh4p is used as a control. It is clearly much more active than ORP1-ORD but this is not commented on. Is ORP-ORD just a poor sterol binder?

This is a good point. According to the results shown in Fig. 1b and Supplementary Fig. 2, ORP1-ORD and Osh4p bind DHE/cholesterol comparably. Thus, we think ORP1-ORD is a good sterol binder, similar to yeast Osh4p. We also summarized the initial velocities of DHE transport by Osh4p and ORP1-ORD in the new Supplementary Fig. 3b. When the acceptor liposomes were free of any PIPs, both ORP1-ORD and Osh4p transport DHE/cholesterol slowly. The DHE transport by ORP1-ORD was enhanced nearly 18-fold when the acceptor liposomes were supplemented with PI(4,5)P₂, while PI4P stimulated DHE transfer by Osh4p less than 9 fold. Therefore, PI(4,5)P₂-stimulation on ORP1 is much more efficient than PI4P-stimulation on Osh4p, and as we stated in the manuscript "ORP1-ORD can efficiently transport cholesterol in the presence of PI(3,4)P₂ or PI(4,5)P₂". We added sentences on page 6 in the revised manuscript.

6. In Fig. 2I and 3j, the assay does not measure transport but only binding or extraction of DHE from the donor liposome; the assay is not designed to register DHE arrival at the acceptor liposome (see #2 above).

Please see our answer to your point #2. The DHE transport assay with DNS-PE incorporated into either donor or acceptor liposomes can be used to measure sterol transfer.

7. In Fig. 4 and extended data 6, it is clearly shown that the ORP1-ORD binds very tightly to PI-3,4-P2 containing liposomes using a sedimentation assay. This being the case, it is highly unlikely that the ORP1-ORD will dissociate significantly from liposomes containing this lipid whether it is bound to sterol or not. There is no evidence provided that PI-3,4-P2 loosens its grip on the ORP1-ORD when sterol is bound. This puts into question the model proposed in which PI-3,4-P2 serves as an allosteric activator.

This is a great point. To examine whether PI(3,4)P₂ will loosen its grip on the ORP1-ORD when sterol is bound, we carried out additional membrane association assay (sedimentation assay) using liposome supplemented with both DHE and PI(3,4)P₂ (0.5% each). The result clearly showed that ORP1-ORD dissociated from the liposomes (new Fig. 8a, last two lanes). Thus, ORP1-ORD can be recruited to liposome/membrane through binding to the PI(3,4)P₂ headgroup, yet ORP1-ORD would eventually extract DHE and dissociate from liposome/membrane. This data corroborated our model that PI(3,4)P₂ serves as an allosteric activator. We added this result on page 16 in the revised manuscript.

8. The authors should show the change of level of PI-3,4-P2 level in the PI3KC2beta knockdown cells, either by mass spec, HPLC or antibody-based methods. Plus, they might try over-expression of PI3KC2beta in HeLa cells and assess its impact on cholesterol transport from LELs.

Thank you for your advice. We have now shown that the level of PI(3,4)P₂ decreased dramatically in PI3KC2beta knockdown cells as indicated by immunofluorescence staining of the lipid using the anti-PI(3,4)P₂ IgG (Echelon Catalog # Z-P034). We have made new Supplementary Fig. 4f and added the methods on pages 31-32 in the revised manuscript.

As suggested, we also tried over-expression of PI3KC2beta in HeLa cells and carried out cholesterol esterification assay to assess its impact on cholesterol egress from LELs. However, the effect of PI3KC2beta overexpression was rather minimal (shown below). This is possibly due to an intact LEL cholesterol trafficking pathway in WT cells, which have high background/basal levels of cholesteryl ester formation. Thus, we put the data here for your information only.

Cholesterol esterification assay (upper panel) and Western blotting analysis (lower panel) of HeLa cells treated with PI3KC2 β siRNA or overexpressing HA-PI3KC2 β .

9. *Related to the model in Fig. 5, what is 'basal extraction' and is it any more efficient than stimulated extraction. If PI-3,4P₂ is a strong binder of the ORP1-ORD, perhaps it is an inhibitor of basal sterol extraction (although Fig. 2i, might suggest otherwise).*

These are great points. The 'basal extraction' in our model is referred to as extraction of cholesterol from membrane in the absence of PI(3,4)P₂ (Fig. 9c). As shown in new Fig. 8a, ORP1-ORD can extract DHE from liposome/membrane either in the absence or presence of PI(3,4)P₂ (please also see answer to your point #7). Thus, the changes in fluorescent signal in the DHE binding assay represent both DHE binding and extraction by ORP1-ORD. We thus performed kinetic analyses to assess the impact of PI(3,4)P₂ on the binding and extraction of a single cholesterol/DHE molecule by ORP1-ORD (Fig. 8b). As discussed in the main text (pages 16-17), when the DHE-containing liposomes were supplemented with PI(3,4)P₂, the K_1 decreased and k_{+2} increased, suggesting that PI(3,4)P₂ can improve both the binding affinity between ORP1-ORD and membrane and the extraction rate of cholesterol from membrane to ORP1-ORD. However, as shown in new Fig. 8c, PI(3,4)P₂ did not affect the amount of DHE extraction. These results demonstrated that PI(3,4)P₂ can enhance the liposome/membrane binding affinity and DHE extraction rate of ORP1-ORD, but had little effect on amount of DHE extraction. Therefore, PI(3,4)P₂ is an allosteric activator (not an inhibitor) of sterol extraction by ORP1.

10. *There is some concern about the nature of the ORP1-ORD used in the study. These truncated versions start at 524 and 534, while ORP1S starts at amino acid 514. While this might be an insignificant structural truncation, there is the possibility*

that removal of these 10-20 amino acids affects the behavior of the ORP1-ORD, perhaps causing the unusual higher order structures described in Fig. 3. The authors also do not address whether ORP1S multimerizes (one would predict this based on their structural data), and it should be easy to test in cells. The authors could have at least commented on this point.

Thank you for your advice. We now analyzed the oligomeric states of ORP1S (514-950) in isolation and in complex with cholesterol or PI(4,5)P₂ by gel filtration chromatography. The ligand-free ORP1S exists in three different states (monomers, dimers and trimers), and binding of cholesterol leads to a shift to monomer regardless of the complex preparation procedures (new Fig. 5a). On the other hand, ORP1S can be largely recruited to the PI(4,5)P₂-incorporated liposomes, but can tetramerize upon free PI(4,5)P₂ binding (new Supplementary Fig. 10b, left panel). The oligomeric states of ORP1S are very similar to that of ORP1-ORD (534-950) (Supplementary Figs. 6a and 10b). Thus, truncation of the N-terminal 20 amino acids would not interfere with the homo-oligomerization property of ORP1S *in vitro* (page 11 in the revised manuscript).

As discussed in the answer to reviewer #1, minor point #1, we carried out additional experiments, showing that ORP1S appeared to form monomer, dimer and trimer *in vivo* (new Fig. 5b).

11. The study is focused on the role of ORP1-ORD in sterol removal from the LEL. This form is very similar to the native ORP1S. However, most studies have reported that ORP1L is strongly localized to LELs and mediates cholesterol export (which they seem to confirm in Fig. 2). The overall tone of the manuscript is that the ORD (ie. ORP1S) is also involved in cholesterol transport from the LEL (the model in Fig5 shows the ORD/ORP1S), but this is not directly addressed in any of the experiments described in Fig. 2 or elsewhere. ORP1S is not known to associate with the LEL, instead it appears to be in the cytoplasm. If their model of LEL targeting by interaction with PI-3,4-P2 is correct, one would assume that ORP1S would be present on the LEL. This could have been addressed by re-expressing ORP1S in the DKO HeLa cells to see if it was sufficient to restore cholesterol trafficking to the ER.

Thank you for pointing this out. We re-expressed the empty vector control, ORP1S, ORP1L or both ORP1S and ORP1L in ORP1 DKO cells and carried out lysosomal cholesterol release assay. It appeared that the expression of either ORP1S or ORP1L could facilitate LEL cholesterol exit in the DKO cells, while a more robust effect was observed in the cells re-expressing both ORP1S and ORP1L (new Supplementary Fig. 4e). We've added these results in the main text (page 7 in the revised manuscript).

12. Lysosomal cholesterol has the potential to activate mTORC1. In ORP1 KO cells, there is more mTORC1 activity as shown in Fig. 2j and k. The authors should check whether deficiency of ORP1 affects autophagy?

Thank you for your advice. We have now checked the effect of ORP1 deficiency on both basal and starvation-induced autophagy. Overall, deficiency of ORP1L alone or both ORP1L/1S impaired basal autophagy as indicated by reduced LC3-II levels in ORP1L KO and ORP1L/1S DKO cells (see below, panel A). Upon starvation, the formation of LC3-II was also downregulated in ORP1L KO cells, suggesting an impairment of autophagy initiation caused by ORP1L deficiency (panel B). This is for your information only as there is no more space in the manuscript to fit in this data.

ORP1L deficiency impaired autophagy. (A) Western blotting analysis of HeLa, ORP1L KO, ORP1L/1S DKO cells treated with or without chloroquine (50 μ M) for 4 h. (B) Western blotting analysis of HeLa and ORP1L KO cells starved in EBSS for 0-4 h.

Minor points:

Figure 3I—“N-terminla” spelled incorrectly

We’ve corrected the spelling.

Figure 4J—not clear which curve refers to the presence or absence of PI45P2

We’ve revised the figure.

Line 69, should say ‘we expressed’ not ‘subcloned’.

This has been revised.

Reviewer #3 (Remarks to the Author):

The non-vesicular lipid transport coupled by phosphoinositide-counter transport are currently a major focus of interest in lipid biology. Many ORPs were suggested to transport sterol or PS against their concentration gradients using a PI4P gradient as an energy source. This work reveals a distinct mechanism from previous findings in that ORP1 mediates a down-hill transport of cholesterol under allosteric regulation of PIPs, which explains the specific role of ORP1 in sterol transport at the LEL and ER interface. The authors unraveled the ill-defined features of ORP1 by various biochemical and cell-biology approaches, which are convincing in overall.

I favor publication of the manuscript in condition that the following points are addressed in the revised version.

We greatly appreciate your positive comments on our work.

ORP1 extracts PI4P and other PIPs well from the liposomes (Fig 1b). However, authors observed that ORP1 does not transport any PIPs at all. This finding is quite intriguing considering the conservation of PI4P transport in other ORPs. This feature needs to be confirmed using the acceptor liposomes containing PS to examine the dependency of membrane association for PIP transport. In addition, DHE transport (Fig 1c) should be tested using the liposomes containing PS in the acceptor liposomes. Another possibility is that for the PI4P transport ORP1 might require the presence of PI(3,5)P₂ or PI(4,5)P₂ in the acceptor membrane, which might help PI4P release by opening the N-terminal lid. Did you examine the PIP₂-dependent transport of PI4P?.

These are great points. As discussed in the answer to reviewer #2, point #1, changes of the fluorescent signal in the assays shown in Fig. 1b and Supplementary Fig. 2a cannot tell whether the lipid is extracted from liposome or not. However, the liposome/membrane association assay shown in Supplementary Fig. 9a can resolve binding/extraction. In fact, PI4P (and DHE/cholesterol) can be extracted by ORP1-ORD while PI(3,4)P₂/PI(4,5)P₂ can recruit ORP1 to liposome/membrane (Fig. 8a).

As suggested, we performed the DHE and PI4P transport assays using acceptor liposomes supplemented with PS as in Fig. 1c, d. As shown below, 4% PS had little effect on the DHE transport by ORP1 (panel a, purple trace). High level of PS (30%) slightly enhanced the DHE transport, yet the stimulation effect was much lower than that of 4% PI(3,4)P₂ (blueish green vs green). This weak PS stimulation on DHE transport may be attributed to PS-mediated membrane association of ORP1 through electrostatic interactions between negatively charged headgroup of PS and basic regions on ORP1-ORD (Fig. 7). As discussed in the answer to reviewer #1 (at the very beginning), though ORP1-ORD can bind and extract PI4P, it cannot transport

PI4P because the ORP1-bound PI4P cannot be released from ORP1-ORD and unloaded to liposome/membrane. Here we show that PS cannot enhance PI4P transport by ORP1, suggesting that PS cannot facilitate the release of PI4P. Therefore, PS (30%) incorporated in the acceptor liposome can slightly enhance DHE transport through membrane targeting, but cannot stimulate PI4P transport.

Transport assays with PS incorporated in acceptor liposome. (a) DHE transport assay. (b) PI4P transport assay.

As to the PIP₂-dependent transport of PI4P, it is troublesome because NBD-PH_{FAPP} was used as the sensor to detect all PIPs in our *in vitro* assays (see answer to reviewer #4, point #1). However, we have demonstrated that PI(3,4)P₂/PI(4,5)P₂ cannot be extracted/transported by ORP1-ORD, yet PI(3,4)P₂/PI(4,5)P₂ could bind and recruit ORP1-ORD to the membranes/liposomes. Thus, we tried to examine the PI4P transport by ORP1-ORD with PI(3,4)P₂/PI(4,5)P₂ incorporated in the donor liposomes either (see below). If PI4P can be transported by ORP1 to the acceptor liposome, the fluorescent signal of NBD would increase. The results indicated that PI(3,4)P₂/PI(4,5)P₂ cannot facilitate the transport of PI4P by

ORP1. These results are for your information only as there is no more space in the manuscript to fit in this data.

**PI4P transport assays with both PI4P and PI(3,4)P₂ or PI(4,5)P₂
embedded in donor liposome.**

*One of the most important data is the crystal structure of PI(4,5)P₂ bound ORP1. The orientation of PI(4,5)P₂ head group is different compared to the PI4P of other ORPs, which was explained by the rotatable bonds in the inositol ring. However, the electron densities of the head group and other region of the ligand are not clearly resolved. The authors should provide unbiased omit maps (*F_o-F_c*) of the PI(4,5)P₂ in several different orientations for the quality check of the ligand structures.*

Thank you for your suggestion. To more clearly resolve the electron density of ligands, we re-processed the raw data and generated omit maps (*F_o-F_c*, contoured at 2.3 σ) of the PI(4,5)P₂ ligands in our PI(4,5)P₂-bound ORP1-ORD structure. As shown in new Supplementary Fig. 10c, the headgroups of two PI(4,5)P₂ ligands are well resolved, and their acyl chains are largely modeled. As we discussed in the main text, PI(3,4)P₂ can also bind to ORP1-ORD because the rotatable property of the PI headgroup. Supplementary Table 1 was revised accordingly.

The binding mode of PI(4)P are well preserved in all reported ORP structures and the strict conservation of the key residues of the head-group binding site suggests PI(4)P binding is a common feature of ORPs. Authors should provide a figure showing the comparison of the PIP binding of ORPs in detail (eg. Structure superposition of PI(4,5)P₂ and PI(4)P including key residues). What is the unique structural features that allow the binding of PI(4,5)P₂ in ORP1 compared to other ORPs?

We agree that PI4P binding is a common feature of ORP/Osh family proteins. Consistently, we also found that ORP1-ORD could bind and extract PI4P (Figs. 1b and 8a). As suggested, we compared the PIP headgroup binding modes in PI(4,5)P₂-ORP1, PI4P-Osh3p, PI4P-Osh4p and PI4P-Osh6p complex structures (shown below, panel a). The PIP headgroup-interacting residues in the core regions of ORP/Osh proteins are strictly conserved (Supplementary Fig. 1, blue triangles), while those from the lid regions are variable (shown below, panel b). The lid of ORP1 does not interact with PI(4,5)P₂, which is likely due to the additional 3/5-phosphate (PI(3,4)P₂/PI(4,5)P₂ vs. PI4P) that may push the lid open as we discussed in the manuscript. Moreover, interacting residues from the core region play key roles in PIP₂ binding, because the core region alone of ORP1 can bind PIP₂ similarly as ORP1-ORD (Supplementary Fig. 7). Therefore, we didn't find unique structural features in ORP1, and we believe that ORP1 can bind PI4P in similar binding mode as Osh proteins.

Comparison of PIP headgroup binding modes. (a) PI(4,5)P₂/PI4P headgroup binding at the tunnel entrance of different ORP/Osh proteins. The dashed lines indicate polar contacts. (b) Sequence alignment of lid regions from different ORP/Osh proteins. Residues participating in PIP headgroup binding were indicated with triangles.

The labels in the figures are too small to read.

Thank you. We've revised the figures accordingly.

Reviewer #4 (Remarks to the Author):

This interesting study focuses on the structure and the function of the lipid transfer protein ORPIL, and in particular its C-terminal part, which corresponds to the ORD domain, (i.e. without the domains known to bind specifically to membranes). The authors suggest that ORPIL is a cholesterol transporter and attempt to demonstrate that it participates in the release of cholesterol from late endosomes/lysosomes (LELs) and to its transfer to the ER. A recent study (Zhao et al. Cell Rep 2017) has already described ORPIL as a lipid transfer protein involved in the exit of chol from LEL, but as a potential chol/PI4P exchanger. However, that study did not provide a consistent mechanism for the PI4P metabolism part. Here the authors suggest that ORP1 does not transfer PIPs, but that PIP2 acts as an allosteric activator to facilitate cholesterol transfer. The authors use in vitro approaches (lipid transfer between liposomes by purified proteins) combined with structural approaches and molecular dynamics simulation. They provide the crystal structure of the ORD in complex with cholesterol or with PI(4,5)P2. In addition, the authors also measure the effect of ORP1/S KO on the distribution and metabolism of intracellular cholesterol, and on the activation of mTORC1, which depends on cholesterol and PI(3,4)P2 levels in LELs, as previously demonstrated. Although the role of this protein seems modest or even dispensable for cell function and LEL function, the study has the merit of providing new, interesting and unexpected results for a protein from this family. Importantly however, some key experiments are not being carried out as they should be, and some controls and information are missing. They are necessary for a good understanding of the manuscript and for the relevance of the study. So I require a major revision for this manuscript. Below are my detailed comments.

Many thanks for your support and careful evaluation.

1) PIP binding/extraction and PIP transfer. The authors make the serious mistake of using the FAPP PH domain as a probe to measure the ORD's binding to PIPs and its PIP transfer activity. It makes no sense to use a probe that only binds specifically to PI(4)P to measure the transport of other PIPs, or the ORD binding to other PIPs, since these other PIPs are not recognized by the FAPP PH domain. The experiments in Fig 1b, 1d, Extended fig 2b, 2d, 5e must be repeated with specific probes for each PIP studied. A biochemical validation of these probes will probably be necessary, if this has not been done previously, before using them for binding or transfer experiments by an ORD (Moser von Filseck et al. 2015 Nat Com has characterized biochemically the FAPP-PH). This will apparently require a lot of work, but it is crucial for this study to demonstrate that the ORD is not able to extract PIPs.

This is an important point. We have carefully characterized NBD-PH_{FAPP}, and it indeed can detect all PIPs *in vitro*, but not DHE and PS (see below). Thus, we used NBD-PH_{FAPP} as a common probe to examine PIP binding or transport. This figure is for your information only as it has been included in our NC paper (Ghai R, *et al. Nat Commun.* 2017).

NBD-PH_{FAPP} selectively recognizes all the PIPs, but not DOPC, PI, DHE and PS.

2) *Activation of mTORC1. A 2-fold increase in S6K phosphorylation is not a “dramatic” increase as written on line 128. It has been shown by Castellano BJ et al. 2017 Science, that in conditions of NPC1 null or inhibited, there is a constitutive activation of mTORC1. Here, the ORP1 KO in the condition without FBS has no visible effect on mTORC1 activation. In addition, to prove that LELs are involved, the authors must manipulate the LDL. Thus, it would be interesting to compare for example the effect of the FBS with that of the lipoprotein-deprived serum. Concerning PI3KC2beta, the paper Marat AL et al. 2017 Science clearly shows a phosphorylation of S6K in conditions without FBS upon PI3KC2beta silencing. Therefore, the authors should tone down their statements on the role of ORP1L in the activation of mTORC1.*

Thank you for your advice. We have toned down in terms of mTORC1 activation caused by ORP1 deficiency. As suggested, we compared the effect of FBS and LDL on mTORC1 activation with that of lipoprotein-deficient serum (LPDS) in HeLa and ORP1L KO cells. As shown below, LPDS induced mTORC1 activation in both HeLa and ORP1L KO cells with the latter showing a slightly stronger effect (lane 2 vs lane 6). The effect of LPDS was augmented when FBS or LDL was added to the media, especially in ORP1L KO cells (lanes 7-8). This is for your information only as there is no more space in the manuscript to fit in this data.

Western blotting analyses of HeLa and ORP1L KO cells grown in LPDS in the absence or presence of FBS or LDL.

3) *There is no evidence in this paper that PI3KC2beta siRNA reduces the level of PI(3,4)P2 in LELs.*

Please see the answer to Reviewer #2, point #8.

4) *I see the arguments suggesting that ORPIL transfers cholesterol from LELs to ER, and not from ER to LELs, but the Extended Data Fig 3f suggest that the ORD sterol transfer rate is very, very slow. Such a weak activity cannot reasonably play an important role in the release of cholesterol from LEL, which is the main entry point for cholesterol into the cell. Moreover, the transfer is a little more effective when the PIPs are in the accepting liposomes (fig 1c). In fact, the results obtained with the ORD domain in Figure 1 do not prove that ORPIL (i.e. the entire form) does the same. For lipid binding and transfer, it has been shown that the interaction of some ORPs, including ORPIL, with VAP influences lipid binding (Weber-Boyvat M et al. CMLS 2015; Mesmin B et al. Cell 2013). The authors should perform similar lipid transfer experiment with the full-length protein ORPIL, and by adding VAP. The membrane binding of the PH domain and the binding to VAP could accelerate the transfer.*

These are great points. Please also see the answer to Reviewer #2, point #5. As we showed in new Supplementary Fig. 3b, the basal DHE transport velocity of ORP1-ORD is ~5-fold lower than that of Osh4p. However, PI(4,5)P₂-stimulation on ORP1 (nearly 18-fold) is much more efficient than PI4P-stimulation on Osh4p (less than 9 fold). Subsequently, the PIP-stimulated sterol transfer rates of ORP1 and

Osh4p are 7 and 17 DHE min^{-1} *per* ORD, respectively. As we stated in the manuscript, ORP1-ORD can efficiently transport cholesterol in the presence of PI(3,4)P₂ or PI(4,5)P₂, and we believe that ORP1L is a *bona fide* cholesterol transporter while we cannot rule out the presence of ORP1L-independent mechanisms. We added sentences on page 6 in the revised manuscript.

As to the mechanism of ORP1L, we agreed that the FFAT motif and/or PH domain might accelerate the transfer of cholesterol by ORP1L. We purified the full-length ORP1L protein and compared the lipid binding profiles of ORP1L and ORP1S/ORP1-ORD. They similarly bind DHE and various PIPs, while ORP1L binds PS weakly which can be attributed to its PH domain. We also examined the DHE transport by ORP1L in the presence of VAP-A. The velocity of basal sterol transfer by ORP1L (w/o PIP) is 3~4-fold higher than ORP1S, which may be owing to the interactions between ORP1L-FFAT and VAP-A which is important for the formation of membrane contact site. When PI(4,5)P₂ or PI(3,4)P₂ was embedded in the donor liposomes, the DHE transport by either ORP1S or ORP1L can be significantly enhanced. However, ORP1L cannot transfer PI4P nor PI(4,5)P₂, regardless of the presence of DHE/cholesterol. Thus, the lipid binding and sterol transfer of ORP1L are similar to that of ORP1S. We made new Fig. 3 and added these results on pages 8-9 in the revised manuscript.

5) Molecular dynamics. It is not clear whether the sterol molecule is in continuous rotation in the ORD tunnel or whether it is transient due to the initial state with the lid open. The question is: does the rotation stop when the lid closes and is it possible to quantify the fluctuations (rotations) of cholesterol in the tunnel over time (ie. during the first 20-30 ns, to do the correlation with the movements of the lid (measured in the Extended Data Fig 5a). This would strengthen the data suggesting that the molecules in the crystal structure are in an intermediate state.

Thank you for your advice. We quantified the rotation of cholesterol by measuring the distance between two atoms throughout the simulation. One atom is the 21st carbon of cholesterol whose position is changed dramatically during MD, and the other is the C α of Arg579 on the core of ORP1-ORD which is very stable during the simulation. We also quantified the movements of the lid by measuring the distance between the C α of Arg554 from the lid and the C α of Arg579. As shown in new Supplementary Fig. 6d, the lid closes the tunnel in 20 ns, while cholesterol reaches a steady state within 5 ns. Thus, we agree that cholesterol in our structure adopts an intermediate state, and added these results on page 12 in the revised manuscript.

6) For the gel filtration, the sentence lines 182-183 is not clear. Moreover, the protocol is also unclear. What are the concentrations of cholesterol and protein used, the incubation conditions? It is also wrong to write that there is no change in the gel filtration profile of fig 3d upon chol addition: The comparison between the black and dark blue lines shows a strong shift towards the monomer. Can the authors provide an explanation?

Thank you for pointing these out. We agreed that it is not appropriate to state that the ligand-free ORP1-ORD exists predominantly in the monomeric form, and have deleted this statement. We also noticed the shift of ORP1-ORD from dimer/trimer to monomer upon cholesterol binding, especially after incubated with cholesterol-containing liposomes. Structural analyses of the ‘closed’ monomeric ORP1-ORD (the MD model) revealed that the lid would make multiple contacts with cholesterol, as that observed in the structures of Osh-cholesterol complexes. That is, the binding of cholesterol would stabilize the monomeric conformation of ORP1-ORD. Furthermore, ORP1-ORD binds to membrane in the monomeric state, with the lid open to allow the extraction or unloading of lipids as shown in our model (Fig. 9c). When dissociated from membrane, the N-terminal loop in the ‘open’ ORP1-ORD more likely binds to the core of its own rather than that of other ORP1-ORD molecules. Therefore, incubation of ORP1-ORD with cholesterol-containing liposomes leads to the obvious shift to monomer (Fig. 5a and Supplementary Fig. 6a). We have added sentences on page 11, and revised the Methods for gel filtration analyses as suggested on page 35 in the revised manuscript.

7) For oligomers, the authors do not discuss their possible physiological relevance, or whether they even exist in the cell. it is not clear if the dimer and trimer (fig 3 e, f) keep the right orientation of the basic patches (all on the same side)? In the tetramer when the ORD is bound to PI(4,5)P2, the lipid appears to be trapped between two proteins, which seems unlikely in reality. Can the authors comment ?

As discussed in the answer to reviewer #1, minor point #1, we carried out additional experiments, showing that ORP1S appeared to form monomer, dimer and trimer in cells (new Fig. 5b).

In the monomeric state, the functional basic patches I and II of ORP1-ORD are in the right orientation, ready to bind simultaneously to the biological membranes enriched in negatively charged lipids. But we don’t think that ORP1-ORD would bind membrane in the dimeric or trimeric states. The lid is also required for membrane association and must be in the ‘open’ conformation to facilitate lipid extraction or unloading. The adjacent N-terminal loop would thus be pulled away from its binding site on the core. We believe membrane targeting would trigger the depolymerization

of dimeric and trimeric ORP1-ORD, and ORP1-ORD binds to membrane in the monomeric, 'open' state.

We have shown that ORP1-ORD can be recruited to the PI(4,5)P₂-incorporated liposomes, but can tetramerize upon free PI(4,5)P₂ binding (Fig. 8a and Supplementary Fig. 10b). As we stated in the main text, the PI(4,5)P₂-bound structure was obtained using the direct mixture of ORP1-ORD and free PI(4,5)P₂ (page 18). Thus, the tetramerization upon free PI(4,5)P₂ binding may not be of physiological relevance, which is consistent with the results that no tetramer was detected in cells (new Fig. 5b). Nevertheless, the polar residues for PI(4,5)P₂ headgroup binding, but not the hydrophobic residues for acyl chain binding, are essential for ORP1 function as a PI(3,4)P₂/PI(4,5)P₂-stimulated cholesterol transporter (Fig. 8d-g). Therefore, as shown in our model (Fig. 9c), we only adopted the information about PI(4,5)P₂ headgroup binding derived from this PI(4,5)P₂-bound structure.

8) Is the fact that the sn-1 chain of PIP2 enters into the ORD tunnel compatible with the main message of the manuscript, ie. that PIP2 stimulates cholesterol transport by the ORD? I could understand if it stimulates the release of cholesterol from the protein by a competition mechanism, but the role of allosteric activator is not very intuitive considering that both ligands occupy the same binding site (at least part of it). An interesting experiment would be to measure the binding of ORD to liposomes containing both DHE and PI(3,4)P2. In this case, is there a reduction in membrane binding compared to when only PI(3,4)P2 is present (fig 4e)? This would help to understand the mechanism. Moreover, the effect of PI4P in this figure is disturbing. The authors discuss little about this effect in the manuscript.

As discussed above in your point #7, we agree that the tetrameric state of ORP1-ORD upon free PI(4,5)P₂ binding seems unlikely in reality. The sole useful information derived from this PI(4,5)P₂-bound structure is the binding mode of PI(4,5)P₂ headgroup, which plays an important role. In our allosteric activation model (Fig. 9c), the acyl chains of PI(3,4)P₂/PI(4,5)P₂ will remain in the membrane, and would not compete with cholesterol.

As suggested, we carried out additional membrane association assay using liposome supplemented with both DHE and PI(3,4)P₂ (new Fig. 8a, last two lanes). The result showed that ORP1-ORD can extract DHE and dissociate from the liposome. As discussed in the answer to Reviewer #2, points #7 & 9, we demonstrated that PI(3,4)P₂ can enhance the liposome/membrane binding affinity and DHE extraction rate of ORP1-ORD, but had little effect on the amount of DHE extraction. Therefore, PI(3,4)P₂ is an allosteric activator of sterol transport by ORP1. We added these results on pages 16-17 in the revised manuscript.

Fig. 8a showed that ORP1-ORD can extract PI4P and then dissociate from liposome/membrane. As we discussed in the answer to reviewer #1 (at the very beginning), ORP1-ORD cannot transport PI4P because the ORP1-bound PI4P cannot be released from ORP1-ORD and unloaded to liposome/membrane. Thus, PI4P may act as an inhibitor in DHE transport by ORP1 in our *in vitro* assay (new Supplementary Fig. 12). We have added discussion about PI4P on page 23 in the revised manuscript.

9) Although the effect on Filipin is clear in Figure 5, how do the authors explain that mutants that are supposed to do less interaction with membranes are much more enriched in LELs? This is particularly striking with mutants of basic patches: they are completely recruited to late endosomes as compared to the WT.

We also noticed that many ORP1L mutants are much more enriched in LELs. We reasoned that ANK domain and/or PH domain also functioned in membrane targeting. When the membrane-binding capacity of ORD was impaired, ANK domain and/or PH domain might function more potently to complement the impairment. In the paper Mesmin *et al. Cell*, 2013, ORD mutants of OSBP also concentrated at Golgi because the mutations promoted the interaction of PH domain with PI4P.

Minor points:

10) Related to Fig 2, the authors should tone down their statements: ORP1L is not "required" for the removal of cholesterol from LELs as it is written line 114. ORP1L has a modest contribution at best (compare the effect of U18666A or NPC1 KO to the effect of the KO of ORP1L and S).

Thanks. We have toned down.

11) In the end of the introduction the authors mention the study by Zhao et al. Cell Rep 2017, but do not mention another study that suggests that in the absence of LDL-cholesterol, ORP1L binds VAP, and this interaction promotes cholesterol transfer from the ER to MVBs to support ILV formation (Eden ER et al. 2016 Dev Cell). So that would work the other way round from the model presented by the authors in the present manuscript. It would be interesting to confront that study with the current one.

This is a good point. ORP1L may facilitate cholesterol transport between LELs and the ER in different directions under different physiological conditions. As shown by Eden ER *et al. (Dev Cell. 2016)*, ORP1L binds VAP and promotes cholesterol

transfer from the ER to MVBs in the absence of LDL-cholesterol. Under such condition, the formation of ILV in MVBs requires cholesterol being delivered from the ER to synthesize new membranes, and hence ORP1L may be involved. In our system, cells were grown in full-serum medium (lipoproteins-rich). Under this condition, cells sort lipoprotein derived cholesterol through LELs, from where free cholesterol is transported to the ER and other parts of the cell for proper functions and growth. As such, we showed in this manuscript that ORP1 plays a role in LELs to ER cholesterol trafficking, which is in agreement with the findings by Zhao *et al.* (*Cell Rep.* 2017). Finally, we have now included the study by Eden ER *et al.* in the Introduction (page 4 in the revised manuscript).

12) Fig 1b shows the % of “accessible” DHE. It should also be written in line 75.

Because of the bulky, polar headgroup, the spontaneous transbilayer flip-flop of phospholipids (such as PIPs and PS) is exceptionally slow with half-times ranging from hours to weeks (Sprong, H. *et al. Nat Rev Mol Cell Biol.* 2001). By contrast, DHE is small and hydrophobic, and easy to flip-flop between the two layers of membrane/liposome. Therefore, the “% accessible” labeled in Fig. 1b is for PIPs and PS, while DHE/cholesterol used in the binding assay is fully accessible to ORP/Osh proteins.

13) In Fig 1, it is sometimes written 460 or 490 for NBD excitation.

The NBD exhibits maximal excitation at 490 nm, which is used in PIP binding assays (Fig. 1b). However, Rhod-PE used in the PIP transport assays (Fig. 1d) will also be excited at 490 nm. Thus, we use 460 nm for NBD excitation to eliminate the disturbance of Rhod-PE in PIP transport assays.

14) Line 107: replace « Extended Data Fig 3c » for « Extended Data Fig 3d ».

Thanks. We've corrected this.

Reviewers' Comments:

Reviewer #1:

Remarks to the Author:

The authors have convincingly addressed my concerns and substantially improved the manuscript.

Reviewer #2:

Remarks to the Author:

The authors have addressed most of my concerns with many new experiments and clarifications. As will any very data-rich manuscript, there are still several outstanding issues mostly related to how PI3,4P2 would promote specificity or enhance transfer rates.

1. The change is acceptable.
2. They have shown that the same pattern of PIP specificity and DHE transfer when DNS-PE is the acceptor liposome. It is noted that with this combination, the level of activation by PIPs is lower (3-fold) and there is higher basal extraction (in the absence of PIPs). Previous papers seem to use DNS-PE in either acceptor or donor liposome.
3. Okay
4. Okay
5. Okay, but as mentioned above, DHE transfer is stimulated less by PIPs when DNS-PE is in the acceptor liposome.
6. Okay (see 2)
7. Explanation is okay but need some clarification on the final two lanes in Fig. 8a, which look like they are from a different experiment (the panel is darker than the other sets of supernatant and pellet fractions, and the loads seem different) and the ratio of intensity does not appear to be only 10% bound.
8. The response is good.
9. Related Fig. 8 B and C, on the surface this seems like a scenario wherein PI3,4P2 stimulates the binding and extraction rate but not the actual rate of sterol-loaded ORP1-ORD release from the membrane. In the context of the model shown in Fig. 9c, the amount of sterol extraction under basal and PI3,4P2 stimulated conditions would be similar, with the main effect of PI3,4P2 being the specific targeting of ORP1-ORD to the membrane; there is no loosening of the PI3,4P2 grip on ORP1-ORD. When looking at the data in Fig. 1 where PI3,4P2 is in the donor membrane and recruiting sterol-loaded ORP1-OHD, in which case the rate enhancement is large, it would appear that the major effect of PI3,4P2 is not sterol binding and extraction but stimulating the release of sterol from ORP1-ORD to the donor membrane. This is not apparent in the model and does not explain how the delivery of sterol is stimulated by PI3,4P2. If off-loading of sterol is also stimulated by PI3,4P2, this suggests a futile cycle.
10. This is now addressed with new experiments using recombinant ORP1S. It was noted in the original manuscript that ORP1S could not be expressed. How was this overcome?
11. The release of LEL cholesterol based on fillipin staining is difficult to see in Sup Fig. 4e. As well this is a very qualitative assay with respect to total cell fluorescent. Would have been better to do a ACAT assay or some other qualitative measure of cholesterol arrival at the ER.
12. Okay, looks like a very mild effect on autophagy.

Some additional points to address:

A. The conditions used for activation of mTORC in Fig. 2g is not appropriate to demonstrate a cholesterol-specific effect. Here they have starved the cells for serum (FBS) and then added FBS back for up to 30 min. The experiments should have involved culturing cells in delipidated serum with or without LDL; otherwise they cannot distinguish between the effects of growth factors and

cholesterol on mTORC activity.

B. A new conclusion from the study is that ORP1-ORD binds and extracts PI4P (shown in Ref. 20) based on FFAP-PH displacement and changes in ORP1-ORD distribution in the sup and pellet fractions of a liposome binding assay. Since they find no evidence of PI4P transfer, the authors posit that it could be an inhibitor but the data shown does not seem to support (or show) this. In fact in Figure 3 C and D it appears to be slightly stimulatory.

C. On Line 480, reference 20 did not measure or propose a PI4P counter current transfer mechanism. That paper reached essentially the same conclusion that you describe on line 472-473, that cholesterol can transport down a concentration gradient to the ER without need for a counter current mechanism. The fact that a PI4P HHA binding mutant of ORP1L was not functional may simply reflect an overall lack of PIP binding either for targeting or transfer.

Reviewer #3:

Remarks to the Author:

I think the authors have addressed most of the reviewer's criticisms properly in the revised manuscript. I recommend the publication of the manuscript.

Reviewer #4:

Remarks to the Author:

In this revised version some ambiguities have been resolved and points being investigated are more directly exposed. The experiments requested have been carried out in most cases. For instance, new in vitro experiments with the long form (ORP1L) and VAP-A are presented - but it is unfortunate that the control without VAP-A is not shown in this context. They also confirmed that multimeric forms of ORP1S were found in vivo, thereby giving a physiological meaning to their structural observation.

Overall, the results of this paper are highly interesting, and unexpected as they are at odds with the lipid exchange mechanism currently accepted for ORPs. The conclusions of this article regarding PI4P remain a little vague though. PI4P can clearly be extracted from the membranes by ORP1, but the irreversible nature of this mechanism seems elusive. However, I acknowledge the effort of the authors to explain this in the discussion. Finally, considering the impressive combination of approaches used, and since this article seems fairly robust on the structure and the function of ORP1, I recommend it for publication.

I have minor points below:

- 1) The method provided for full length ORP1L purification is not enough detailed: Please indicate the steps of cell lysis with the reagents used before the column.
- 2) Figure 8a: specify that the last two lanes are not from the same gel. The best way is to draw a dashed line between the first part of the gel and these last two lanes and mention it in the figure legend.
- 3) The sentence line 169 to 171 is not clear. Please rewrite it.
- 4) The description of DHE should appear on line 81 when it is first mentioned.
- 5) A typo in legend fig 4 line 31: U1866A-mediated instead of U18666A-mediated.

Reviewers' comments:

Reviewer #1 (Remarks to the Author):

The authors have convincingly addressed my concerns and substantially improved the manuscript.

Reviewer #2 (Remarks to the Author):

The authors have addressed most of my concerns with many new experiments and clarifications. As will any very data-rich manuscript, there are still several outstanding issues mostly related to how PI3,4P2 would promote specificity or enhance transfer rates.

1. The change is acceptable.

2. They have shown that the same pattern of PIP specificity and DHE transfer when DNS-PE is the acceptor liposome. It is noted that with this combination, the level of activation by PIPs is lower (3-fold) and there is higher basal extraction (in the absence of PIPs). Previous papers seem to use DNS-PE in either acceptor or donor liposome.

Response: We noticed that the basal extraction (in the absence of PIPs) is much higher in the DHE transport assay with DNS-PE incorporated into the acceptor liposomes, which results in a lower level of PI(3,4)P₂/PI(4,5)P₂ activation. We also examined the PI4P-stimulation on sterol transport by Osh4p, which displayed a 3.5-fold activation (see below), much lower than that (8.7-fold) in the other assay with DNS-PE embedded in the donor liposome (Fig. 1c and Supplementary Fig. 3b). Importantly, in both sterol transport assays, PI(3,4)P₂ and PI(4,5)P₂ are the only two PIPs that can markedly stimulate the DHE transport by ORP1.

DHE transport assay for Osh4p with DNS-PE incorporated in the acceptor liposome. The donor liposome (L_A , DOPC/DOPE, 2:1; total lipids, 200 μ M) was doped with 5% DHE, and the acceptor liposome (L_B , DOPC/DOPE, 2:1; total lipids, 200 μ M) contained 2.5% DNS-PE and 4% PI4P. The DHE transport was followed by measuring the dansyl signal at 525 nm (bandwidth 10 nm) upon DHE excitation at 310 nm (bandwidth 10 nm).

3. Okay

4. Okay

5. Okay, but as mentioned above, DHE transfer is stimulated less by PIPs when DNS-PE is in the acceptor liposome.

Response: See answer to your point #2.

6. Okay (see 2)

7. Explanation is okay but need some clarification on the final two lanes in Fig. 8a, which look like they are from a different experiment (the panel is darker than the other sets of supernatant and pellet fractions, and the loads seem different) and the ratio of intensity does not appear to be only 10% bound.

Response: Thank you for pointing this out. The final two lanes are from a different experiment (new Supplementary Fig. 9b). Similarly, liposomes supplemented with 0.5% PI(3,4)P₂ recruited >90% proteins, while >80% DHE molecules were extracted. When the liposomes were doped with 0.5% PI(3,4)P₂ and increasing amount (0.1-1.0%) of DHE, PI(3,4)P₂ indeed loosens its grip on the ORP1-ORD. For comparison, we put the result with DHE and PI(3,4)P₂ both at 0.5% level beside the data with only one type of indicated lipids (Fig. 8a), and mention this difference in the figure legend as suggested by Reviewer #4. We also drew a dashed line between the first part of the gel and the last two lanes, and the percentage of protein bound on liposomes calculated from the last two lanes was corrected.

8. The response is good.

9. Related Fig. 8 B and C, on the surface this seems like a scenario wherein PI3,4P₂ stimulates the binding and extraction rate but not the actual rate of sterol-loaded ORP1-ORD release from the membrane. In the context of the model shown in Fig. 9c, the amount of sterol extraction under basal and PI3,4P₂ stimulated conditions would be similar, with the main effect of PI3,4P₂ being the specific targeting of ORP1-ORD to the membrane; there is no loosening of the PI3,4P₂

grip on ORP1-ORD. When looking at the data in Fig. 1 where PI3,4P2 is in the donor membrane and recruiting sterol-loaded ORP1-OHD, in which case the rate enhancement is large, it would appear that the major effect of PI3,4P2 is not sterol binding and extraction but stimulating the release of sterol from ORP1-ORD to the donor membrane. This is not apparent in the model and does not explain how the delivery of sterol is stimulated by PI3,4P2. If off-loading of sterol is also stimulated by PI3,4P2, this suggests a futile cycle.

Response: The cell biological data showed that ORP1 regulates lysosomal cholesterol homeostasis with PI(3,4)P₂ at the same limiting membrane of LELs, and we thereby focused on the mechanism how PIPs can regulate the transport of cholesterol on the same liposome. In our scenario, PI(3,4)P₂ can improve both the binding affinity between ORP1-ORD and membrane and the extraction rate of cholesterol from membrane to ORP1-ORD, but had little effect on the amount of DHE extraction (Figs. 8b, c and 9c). We agree that this allosteric regulation mechanism might be used in the sterol unloading/release process as well (Fig. 1c). Note that such an allosteric activation mechanism by PI(3,4)P₂/PI(4,5)P₂ can only be applied when the cholesterol transport is “downhill” from LELs (high cholesterol) to the ER (low cholesterol). However, we don’t think this will lead to a futile cycle as PI(3,4)P₂ can improve the extraction (and unloading) rate when there is sterol concentration gradient. Indeed, in a test with PI(3,4)P₂ embedded in both donor and acceptor liposomes, the stimulation of DHE transport was observed as well (see below).

DHE transport assay of ORP1-ORD with PI(3,4)P₂ embedded in the donor and/or acceptor liposomes.

10. This is now addressed with new experiments using recombinant ORP1S. It was noted in the original manuscript that ORP1S could not be expressed. How was this overcome?

Response: This is a misunderstanding. We have stated in the original manuscript, “.....a short isoform ORP1S (residues 514-950) containing only the ORD (Fig. 1a, Extended Data Fig. 1). We subcloned the ORP1S and a series of N-terminal deletion variants, among which only ORP1S and two truncations (residues 524-950 and 534-950) were soluble.”

11. The release of LEL cholesterol based on fillipin staining is difficult to see in Sup Fig. 4e. As

well this is a very qualitative assay with respect to total cell fluorescent. Would have been better to do a ACAT assay or some other qualitative measure of cholesterol arrival at the ER.

Response: Given the variations of transfection efficiencies of different plasmids, as well as the levels of transiently expressed proteins in HeLa cells, we have always found that the ACAT assays combined with transient overexpression were problematic, sometimes producing inconsistent results. Filipin staining experiments as shown in Supplementary Fig. 4e, on the other hand, could specifically examine transfected cells and reveal the intensity of LEL cholesterol in these cells. By quantifying the fluorescence of filipin, at least a semi-quantitative measurement can be obtained to indicate the extent of LEL cholesterol release to the ER (e.g. Figs. 2d and 9b). We have now also quantified filipin intensities from the images shown in Supplementary Fig. 4e and included the results as new Supplementary Fig. 4f. The quantitation data showed that the expression of either ORP1S or ORP1L, or both ORP1S and ORP1L, could significantly facilitate LEL cholesterol exit in the DKO cells.

12. Okay, looks like a very mild effect on autophagy.

Some additional points to address:

A. The conditions used for activation of mTORC in Fig. 2g is not appropriate to demonstrate a cholesterol-specific effect. Here they have starved the cells for serum (FBS) and then added FBS back for up to 30 min. The experiments should have involved culturing cells in delipidated serum with or without LDL; otherwise they cannot distinguish between the effects of growth factors and cholesterol on mTORC activity.

Response: We have addressed this point when responding to the concerns from Reviewer #4, point 2. When cultured in LPDS medium plus LDL, ORP1L KO cells showed increased mTORC1 activity compared with those cells cultured in LPDS medium. Please refer to the figure (lane 8 vs lane 6) in the answer to Reviewer #4, point 2 in last rebuttal.

Response: *Thank you for your advice. We have toned down in terms of mTORC1 activation caused by ORP1 deficiency. As suggested, we compared the effect of FBS and LDL on mTORC1 activation with that of lipoprotein-deficient serum (LPDS) in HeLa and ORP1L KO cells. As shown below, LPDS induced mTORC1 activation in both HeLa and ORP1L KO cells with the latter showing a slightly stronger effect (lane 2 vs lane 6). The effect of LPDS was augmented when FBS or LDL was added to the media, especially in ORP1L KO cells (lanes 7-8). This is for your information only as there is no more space in the manuscript to fit in this data.*

Western blotting analyses of HeLa and ORP1L KO cells grown in LPDS in the absence or presence of FBS or LDL.

B. A new conclusion from the study is that ORP1-ORD binds and extracts PI4P (shown in Ref. 20) based on FFAP-PH displacement and changes in ORP1-ORD distribution in the sup and pellet fractions of a liposome binding assay. Since they find no evidence of PI4P transfer, the authors posit that it could be an inhibitor but the data shown does not seem to support (or show) this. In fact in Figure 3 C and D it appears to be slightly stimulatory.

Response: We have addressed this point in last rebuttal when responding to the concerns from reviewer #1 (at the very beginning). The concentration of ORP1 (FL or ORD) used in the DHE transport assays is 0.1 μM , which allows us to show the PI(4,5)P₂/PI(3,4)P₂-stimulation on the initial velocity of DHE transfer (Figs. 1c and 3c, d). The effects of other PIPs were not clear. To examine the effect of PI4P, we increased the concentration of ORP1-ORD to 0.5 μM , and PI4P did inhibit DHE transfer by ORP1 (Supplementary Fig. 12b).

C. On Line 480, reference 20 did not measure or propose a PI4P counter current transfer mechanism. That paper reached essentially the same conclusion that you describe on line 472-473, that cholesterol can transport down a concentration gradient to the ER without need for a counter current mechanism. The fact that a PI4P HHAA binding mutant of ORP1L was not functional may simply reflect an overall lack of PIP binding either for targeting or transfer.

Response: Thank you for pointing this out. We agree that Ref. 20 (Zhao and Ridgeway, *Cell Reports*, 2017) did not propose a PI4P counter transfer mechanism, but PI4P binding activity of ORP1L has been shown to be required for cholesterol removal from LELs. We have made revision on pages 4 and 23.

Reviewer #3 (Remarks to the Author):

I think the authors have addressed most of the reviewer's criticisms properly in the revised manuscript. I recommend the publication of the manuscript.

Reviewer #4 (Remarks to the Author):

In this revised version some ambiguities have been resolved and points being investigated are more directly exposed. The experiments requested have been carried out in most cases. For instance, new in vitro experiments with the long form (ORP1L) and VAP-A are presented - but it is unfortunate that the control without VAP-A is not shown in this context. They also confirmed that multimeric forms of ORP1S were found in vivo, thereby giving a physiological meaning to their structural observation.

Overall, the results of this paper are highly interesting, and unexpected as they are at odds with the lipid exchange mechanism currently accepted for ORPs. The conclusions of this article regarding PI4P remain a little vague though. PI4P can clearly be extracted from the membranes by ORP1, but the irreversible nature of this mechanism seems elusive. However, I acknowledge the effort of the authors to explain this in the discussion. Finally, considering the impressive combination of approaches used, and since this article seems fairly robust on the structure and the function of ORP1, I recommend it for publication.

Response: Thank you for your positive comments. Shown below is the control data without VAP-A, which had little effect on the basal activity of ORP1L. For better comparison of data shown in Fig. 3c, d, this control was not included in the main text.

DHE transport assay of ORP1L with or without VAP-A and PI(3,4)P₂ embedded in the donor liposomes.

I have minor points below:

1) The method provided for full length ORP1L purification is not enough detailed: Please indicate the steps of cell lysis with the reagents used before the column.

Response: Thank you for pointing this out. We've added the lysis step for the preparation of full-length ORP1L in the Methods section.

2) Figure 8a: specify that the last two lanes are not from the same gel. The best way is to draw a dashed line between the first part of the gel and these last two lanes and mention it in the figure legend.

Response: Thank you for your suggestion. We drew a dashed line between the first part of the gel and these last two lanes and mentioned the difference in the figure legend. We also show results of the additional membrane association assays using liposomes doped with 0.5% PI(3,4)P₂ and increasing amount (0.1-1.0%) of DHE in the new Supplementary Fig. 9b.

3) The sentence line 169 to 171 is not clear. Please rewrite it.

Response: The sentence was revised as “*The FFAT motif of ORP1L can interact with VAP-A, which is important for the formation of membrane contact site^{26, 27}. Consistently, the initial velocity of basal sterol transfer (w/o PIP) by the full-length ORP1L is 3~4-fold higher than that by the ORD alone (Fig.3c, d).*”.

4) The description of DHE should appear on line 81 when it is first mentioned.

Response: Thank you for pointing this out. We've revised it.

5) A typo in legend fig 4 line 31: U1866A-mediated instead of U18666A-meidated.

Response: We've corrected it.

Reviewers' Comments:

Reviewer #2:

Remarks to the Author:

nice work--interesting study

all my concerns are addressed